# Potential of Dietary HDAC2i in Breast Cancer Patients Receiving PD-1/PD-L1 Inhibitors

**DOI:** 10.3390/nu15183984

**Published:** 2023-09-14

**Authors:** Yuqian Wang, Lingeng Lu, Changquan Ling, Ping Zhang, Rui Han

**Affiliations:** 1Department of Chinese Medicine Oncology, The First Affiliated Hospital of Naval Medical University, Shanghai 200433, China; 2Department of Chinese Medicine, Naval Medical University, Shanghai 200433, China; 3Department of Chronic Disease Epidemiology, Yale School of Public Health, Yale University, 60 College Street, New Haven, CT 06520, USA; 4School of Medicine, Center for Biomedical Data Science, Yale University, 60 College Street, New Haven, CT 06520, USA; 5Yale Cancer Center, Yale University, 60 College Street, New Haven, CT 06520, USA; 6Center for Integrative Conservation, Yunnan Key Laboratory for the Conservation of Tropical Rainforests and Asian Elephants, Xishuangbanna Tropical Botanical Garden, Xishuangbanna 666303, China; 7Department of Oncology, The First Hospital Affiliated to Guangzhou University of Chinese Medicine, Guangzhou 510405, China

**Keywords:** dietotherapy, breast carcinoma, HDAC2 suppression, immune checkpoint inhibitor, immunotherapy sensitizer

## Abstract

Breast cancer (BC) is a lethal malignancy with high morbidity and mortality but lacks effective treatments thus far. Despite the introduction of immune checkpoint inhibitors (ICIs) (including PD-1/PD-L1 inhibitors), durable and optimal clinical benefits still remain elusive for a considerable number of BC patients. To break through such a dilemma, novel ICI-based combination therapy has been explored for enhancing the therapeutic effect. Recent evidence has just pointed out that the HDAC2 inhibitor (HDAC2i), which has been proven to exhibit an anti-cancer effect, can act as a sensitizer for ICIs therapy. Simultaneously, dietary intervention, as a crucial supportive therapy, has been reported to provide ingredients containing HDAC2 inhibitory activity. Thus, the novel integration of dietary intervention with ICIs therapy may offer promising possibilities for improving treatment outcomes. In this study, we first conducted the differential expression and prognostic analyses of HDAC2 and BC patients using the GENT2 and Kaplan–Meier plotter platform. Then, we summarized the potential diet candidates for such an integrated therapeutic strategy. This article not only provides a whole new therapeutic strategy for an HDAC2i-containing diet combined with PD-1/PD-L1 inhibitors for BC treatment, but also aims to ignite enthusiasm for exploring this field.

## 1. Introduction

Breast cancer is one of the most common malignancies in women worldwide, which can occur in both men and women, and ranks as the second cancer-related cause of death worldwide [1,2]. Its development has been reported to be significantly associated with dietary habits (such as alcohol consumption, high intake of total fat, and low consumption of dietary fiber, etc.) [3]. Five molecular subtypes are classified based on the expression levels of the estrogen receptor (ER), progestogen receptor (PR), human epidermal growth factor receptor 2 (HER2) and Ki-67: Luminal A (ER positive and/or PR positive, HER2 negative, Ki-67 < 20%); Luminal B (HER2 negative/B1: ER positive and/or PR < 20%, HER2 negative, Ki-67 ≥ 20%; HER2 positive/B2: ER positive and/or PR positive, HER2 overexpression); HER2 positive type (HER2 positive, ER negative, PR negative); triple-negative (HER2 negative, ER negative, and PR negative) (TNBC); and other special types [1,2]. The recurrence and/or metastasis of breast cancer significantly contributes to breast cancer-specific mortality, as recurrent tumors tend to be more aggressive and often show resistance to currently available treatments [4]. Thus, to improve treatment outcomes, including overall survival (OS) and disease control, endocrine-, chemo-, targeted- and immuno-therapy alone or in combination have been studied extensively [5]. Immune checkpoint inhibitors (ICIs), which reactivate exhausted CD8+ T cells to kill tumor cells [6], have especially made significant progress in the clinical application of breast cancer treatment [5]. However, the benefit population of breast cancer from ICI monotherapy is limited, making the combined regimen a new research hotspot [5].

The usage of ICIs is to harness the body’s immune system to recognize and attack cancer cells more effectively. For instance, by blocking the interaction between immune checkpoints, such as programmed cell death protein 1 (PD-1) and its ligand (PD-L1), the “brakes” on the immune system could be released, allowing immune effector cells to recognize and attack cancer cells more effectively [5]. Fortunately, ICIs have demonstrated a certain efficacy in combination with chemotherapy in the treatment of both early- and late-stage triple-negative breast cancer (TNBC) [7]. The KEYNOTE-355 trial evaluated the efficacy of PD-1 antibody pembrolizumab in combination with chemotherapy for patients with metastatic TNBC. In the context of early-stage disease, the KEYNOTE-522 study demonstrated significant benefits by adding pembrolizumab to chemotherapy, irrespective of the PD-L1 status [7,8,9]. Furthermore, ongoing studies are currently exploring the application of ICI therapy for hormone receptor-positive and HER2-positive breast cancer [7]. Additionally, novel ICIs are being investigated for their potential across all breast cancer subtypes [7].

Despite these advancements, there are still important unresolved issues. For instance, determining the optimal partners for ICIs to mitigate ICI side effects, and predictive biomarkers to identify who benefit from ICI therapy [7,10]. Another issue that cannot be ignored is that ICIs caused loss of appetite, consequently reducing the patient’s nutritional intake [11], and aggravating the patient’s already damaged physique and immunity, forming a vicious circle.

Histone deacetylases 2 (HDAC2), as a member of HDACs, is a special protease involved in the tightening of the chromatin structure and suppression of gene transcription. Its expression level was significantly in positive correlation with the poor overall survival of patients with BC or hepatocellular carcinoma (HCC) and with the potential adverse effect of patients to PD-1 antibody therapy [12,13,14,15]. Moreover, breast cancer tissues show significantly higher HDAC2 expression than normal breast tissue [13,14,15]. HDAC2 inhibition is a potential anti-cancer agent against breast cancer [16,17,18]. HDAC2 inhibitors can exert a synergistic effect with ICIs [18,19]. Given that short-chain fatty acids (SCFA, e.g., butyric acid) are the fermentation products of dietary fibers, dietary intervention may provide a superior safety profile improving immunotherapy by functioning as HDAC2i [20]. Furthermore, SCFA-producer *Lachnoclostridium* in tumors was positively associated with infiltrating CD8+ T cells and chemokines of CXCL9 and CXCL10, as well as better survival in melanoma cancer [21].

Healthy eating is recommended for cancer patient survivors according to guidelines. Besides providing nutrition and energy intake, dietary intervention has been found to improve the body’s immunity and anti-cancer activity [22]. Therefore, based on our previous study and the latest findings, this review, for the first time, brings out a novel dietary intervention with potential synergistic effects for ICIs therapy by adding HDAC2i containing food, aiming to ignite enthusiasm to explore the potential application of an HDAC2i-containing diet combined with ICI for BC patients, and to eventually improve the clinical benefits for patients.

## 2. Materials and Methods

A comprehensive search of the literature was conducted to identify relevant reports from 2000 to 2023, using the PubMed and Web of Science databases. The search utilized various keywords, either individually or in combinations, including PD1, PDL1, breast cancer, HDAC2 inhibitor, immune checkpoint inhibitors, dietary intervention, herbal remedy, neoadjuvant, and immunotherapy sensitizer. The identified articles were thoroughly reviewed, with preference given to the most recent ones. Additional sources were extracted from the reference lists of selected papers. Priority was primarily assigned to original research and review articles based on animal studies and clinical trials.

Differential expression and prognostic analysis. The differential expression analysis of HDAC2 across cancer and normal tissues were obtained from the GENT2 platform (http://gent2.appex.kr/gent2/, accessed on 11 July 2023). The prognostic analysis was performed by applying the Kaplan–Meier plotter platform (https://kmplot.com/analysis/index.php?p=background, accessed on 11 July 2023). BC patients with high or low HDAC2 protein (or gene) expression were divided by median expression level or by best cutoff value. Cutoff values used in analysis for HDAC2 gene and protein were 1568 and 3, respectively. The total amount of samples for analyzing the HDAC2 gene figure was 4929 (high 2465 vs. low 2464), its counterpart for HDAC2 protein was 65 (high 47 vs. low 18).

## 3. Results

The HDAC2 expression profile across cancer and normal tissues was detected by using the GENT2 platform. Significant difference in HDAC2 expression was displayed between breast cancer tissue and normal breast tissue (*p* < 0.001, Log2FC, 0.122) (Figure 1A). (Significant test results by Two-sample *t*-test for the HDAC2 expression profile is displayed in Appendix A). As shown in Kaplan–Meier survival curves (Figure 1B,C), BC Patients with a high expression of the HDAC2 gene [hazard ratio (HR), 1.65; *p* < 1 × 10^−16^; Figure 1B] or high expression level of the HDAC2 protein (HR, 2; *p* = 0.059; Figure 1C) exhibited a less favorable prognosis compared with patients with low expression levels. However, a significant difference has not been detected in the group of HDAC2 proteins.

## 4. Discussion

### 4.1. HDAC2: A Potential Index of Aggressiveness and a Therapeutic Target against BC

HDAC2 is an enzyme involved in the tightening of the chromatin structure and suppression of gene transcription. Evidence has indicated that HDAC2 is overexpressed in breast cancer cells compared to normal breast tissue. Higher levels of HDAC2 have also been associated with more aggressive tumor characteristics, such as increased cell proliferation, invasion, and metastasis [12,13,14,15]. For instance, a clinical study that included 226 BC patients reported that expression of HDAC2 protein is significantly higher in breast cancer than in benign tumors and indicates that HDAC2 may be involved in invasion, metastasis, anthracyclines therapy resistance, and poor prognosis of sporadic breast cancer (Table 1) [14]. Another study, conducted by Afroditi Nonni’s group, also examined the expression of HDAC2 in 118 deceased sporadic BC patients and its correlation with clinicopathological characteristics of the tumor and the prognosis of the patient. It was found that high expression of the HDAC2 protein is associated with higher histological grade, stage of disease, and worse prognosis (Table 1) [12]. As for the expression of the HDAC2 gene, evidence has also reported that HDAC2 is one of the most commonly amplified genes in aggressive basal-like breast cancer. Additionally, overexpression of HDAC2 was significantly correlated with high tumor grade, positive lymph node status, and poor prognosis [13]. Those results are also consistent with the outcomes of our analysis. For instance, our results indicated that expression of HDAC2 in BC tissue was significantly higher than that in breast normal tissue (*p* < 0.001) (Figure 1). Moreover, the trend can be observed that BC patients with high expression of HDAC2 gained worse prognosis than those without (Figure 1C). However, a significant difference was been detected (*p* = 0.059) (Figure 1). We speculated that the reason may be due to an insufficient sample size (n = 75).

### 4.2. HDAC2 Inhibition for Treating Breast Cancer

HDAC2 has been identified as a crucial regulator of epigenetic control in BC and HDAC2 suppression has been further proved to be an effective approach to treating BC by numerous studies [24,25,26]. For instance, HDAC2 inhibition has been observed to inhibit cellular proliferation in a p53-dependent manner in BC cells [27]. MiR-155 can also decrease the expression of erythroblastic oncogene B by targeting HDAC2 [28]. Moreover, evidence has indicated that PELP1 (proline, glutamate, and leucine-rich protein 1) can bind to miR-200a and miR-141 promoter sequences and modulate the expression of these miRNAs by recruiting HDAC2; therefore, regulating tumorigenic and metastatic potential of BC cells [29]. Thus, a molecular network involving HDAC2 has been considered to serve as a target for developing anti-cancer drugs [29].

Inhibition of HDAC2 has also been found to exert a synergistic effect in BC treatment. For instance, evidence indicates that depleting HDAC2 can sensitize breast cancer cells to apoptosis induced by epirubicin. This finding highlighted the potential of HDAC2 as a therapeutic target and a biomarker in treating breast cancer [30]. Moreover, the modulation of estrogen receptor (ER) signaling is a promising therapeutic approach in ER-expressing breast cancers, and the progesterone receptor (PR) also plays a critical role in this process [31]. Interestingly, selective inhibition of HDAC2, has been found to enhance the apoptotic effects of tamoxifen (a commonly used drug for ER/PR-positive breast cancer) and has demonstrated significant antitumor activity [31]. Additionally, HDAC2 inhibition has also been found to strongly restrain the multidrug-resistance of BC cells induced by the SRGN (serglycin)–YAP (YES-associated protein) axis, therefore enhancing the therapeutic effect of chemotherapy [32].

Taken together, these findings suggested that HDAC2-targeting intervention could represent an effective approach for breast cancer control.

### 4.3. HDAC2 Inhibition Enhances the Therapeutic Effect of ICIs in BC Treatment

The emergence of immune checkpoint inhibitors (ICI) has revolutionized the treatment of breast cancer [33,34,35,36,37,38,39,40]. Adjuvant or neoadjuvant immune checkpoint blockades are used for metastatic breast cancer [34]. Despite ICI therapy being promising, breast cancer cells often find ways to evade the host’s immune system, necessitating combination therapies to overcome these limitations. HDAC inhibitors (HDACi) have demonstrated potent immunomodulatory activity, making them a rational choice for cancer immunotherapies.

#### 4.3.1. HDAC2 Regulates PD-L1 Nuclear Translocation

As a ligand of PD-1, high PD-L1 levels indicate tumor progression and are associated with poor prognosis in immunotherapy-treated human cancer [35]. PD-L1 nuclear translocation has been identified as a key mechanism underlying the immune evasion of BC cells, hindering PD-1 inhibitors [36]. Both cytoplasmic and nuclear PD-L1 can exert immunosuppressive functions on BC cells [37]. Nuclear PD-L1 has been linked to various cellular processes, for example, increasing the anti-apoptotic capacity of tumor cells, promoting mTOR activity, and upregulating glycolytic metabolism [36]. A study has shown that the level of nuclear PD-L1 expression was positively correlated with immune response-related transcription factors, such as STAT3, RelA (p65), and c-Jun [19]. Nuclear PD-L1 interacts with transcription factors, such as RelA and the IFN regulatory factor (IRF), influencing antitumor immunity. Inhibition of nuclear PD-L1 expression led to downregulation of genes involved in evading immune surveillance, such as PDCD1LG2 (encoding PD-L2), VSIR (encoding VISTA), and CD276 (encoding B7-H3), which enhance cytotoxic T-lymphocyte depletion and promote tumor aggressiveness, distant metastasis, and resistance to PD-L1/PD-1 blockade therapy [18,19,38] (Figure 2).

Several recent studies have reported that HDAC2-associated deacetylation promotes the nuclear translocation of PD-L1, leading to tumor immune evasion in BC cells [18,19]. HDAC2 decreases the acetylation level of Lys 263 in the C-tail of PD-L1, resulting in PD-L1 nuclear translocation. In contrast, depletion of endogenous HDAC2 using siRNA, shRNA, or CRISPR-Cas9 increases PD-L1 acetylation [19]. Selective HDAC2is, such as Santacruzamate A (SCA) and ACY957, increase the acetylation of PD-L1, blocking the PD-L1 nuclear translocation. Clathrin-dependent endocytosis is involved in PD-L1 nuclear translocation, and HIP1R initiates this process by interacting with PD-L1’s C-tail. Lys 263 acetylation directly blocks the interaction between HIP1R and PD-L1. Overall, HDAC2 inhibition disrupts PD-L1 nuclear translocation, potentially enhancing the therapeutic efficacy of immune checkpoint inhibitors and boosting antitumor immune responses for BC [19].

#### 4.3.2. HDAC2 Regulates IFN-γ- Induced PD-L1 Expression

a.IFN-γ upregulates the expression of PD-L1

Evidence has revealed that high expression of PD-L1 is correlated with advanced histology and lymph node metastasis in TNBC and HER2+ subtypes, indicating a poor prognosis biomarker [39]. Moreover, PD-L1 knockdown has been found to inhibit the proliferation and migration of TNBC cells [40]. To date, the clinical trials of immunotherapies based on the PD-1/PD-L1 antagonists have shown a notable and durable response in TNBC patients, indicating PD-L1 as a crucial therapeutic target [41,42].

Interferon-γ (IFN-γ) is a crucial cytokine in both innate and adaptive immunity and should be considered as an important driving force for PD-L1 expression in tumor microenvironment. It is able to induce PD-L1 expression on BC cells and increase the apoptosis of antigen-specific T cells, such a process is referred to as “adaptive resistance” [43]. Depleting IFN-γ receptor 1 has been reported to decrease the expression of PD-L1 expression in BC cells, increase the amount of tumor-infiltrating CD8+ lymphocytes or CTLs, and to inhibit the development of BC cells [44]. In addition, injection of IFN-γ into subcutaneous BC cells induced PD-L1 expression and promoted the growth of BC cells. Inversely, PD-L1 depletion completely abrogated the growth of BC cells induced by IFN-γ injection [44] (Figure 2).

b.HDAC2i affects IFN-γ induced PD-L1

HDAC2 can promote PD-L1 induced via IFN-γ stimulation in BC cells [18,19]. Specifically, IFN-γ can induce gene transcription involving STAT1 binding to the gamma interferon activation site (GAS), recruiting HAT and HDAC for chromatin remodeling [45]. Following IFN-γ stimulation, BRD4 is rapidly recruited to the PD-L1 locus, accompanied by increased H3K27ac and RNA Polymerase II (RNA Pol II) occupancy in cancer cells [45]. A ChIP-qPCR assay confirmed enhanced HDAC2 binding to the PD-L1 promoter after IFN-γ treatment. HDAC2 knockout led to reduced STAT1 occupancy and bromodomain-containing 4 (BRD4) recruitment to the PD-L1 promoter, attenuated H3K27ac and H3K9ac (markers of active transcription in the PD-L1 promoter) upregulation induced by IFN-γ, highlighting HDAC2’s role in activating PD-L1 expression through IFN-γ induced signaling pathways [18]. Moreover, upon the binding of IFN-γ to interferon receptors, it transduces signal through Janus kinases (JAKs), signal transducer and activators of transcriptions (STATs) [36]. The JAK/STAT1 pathway activated by IFN-γ positively correlates with PD-L1 expression and plays a critical role in breast cancer immune escape [46]. IFN-γ treatment induces phosphorylation of JAK1, JAK2, and STAT1 in TNBC cells, leading to upregulated STAT1 and PD-L1 expression. However, HDAC2 knockdown inhibits the phosphorylation of JAK1, JAK2, and STAT1, resulting in decreased IFN-γ-induced PD-L1 expression on the TNBC cell surface. HDAC2 knockout also hinders the translocation of STAT1 to the nucleus and inhibits intracellular PD-L1 expression stimulated by IFN-γ, indicating HDAC2’s promotion of IFN-γ induced PD-L1 expression in TNBC cells via JAK-STAT1 pathway activation [18] (Figure 2).

### 4.4. Dietary Intervention Is Important for Breast Cancer Patients Receiving Anti-Cancer Immunotherapy

A healthy diet rich in vitamins, minerals, antioxidants, and phytochemicals, provides essential ingredients for building and strengthening a healthy immune system. Thus, the American Cancer Society (ACS) releases the Nutrition and Physical Activity Guideline for Cancer Survivors, which enhances immune function by supporting essential nutrients [47,48,49,50], helping to optimize the effectiveness of immunotherapy and improve treatment outcomes. Moreover, anti-cancer immunotherapy also possesses toxicities and can lead to various adverse effects on breast cancer patients, such as fatigue, nausea, loss of appetite, and gastrointestinal issues [51]. However, proper dietary interventions have been found to help alleviate these side effects by providing proper nutrition, promoting hydration, and supporting gastrointestinal health [52,53,54]. In addition (Reducing Inflammation), chronic inflammation has been found to be associated with cancer development and progression. Certain foods, such as fruits, vegetables, whole grains, and omega-3 fatty acids, possess anti-inflammatory properties [55,56,57]. Including these foods in the diet can help reduce inflammation, potentially benefiting breast cancer patients undergoing immunotherapy.

Furthermore (Maintaining Weight and Nutritional Status), cancer and its treatment can lead to weight loss, malnutrition, and muscle wasting, which can further weaken the body and hinder treatment effectiveness [58]. Dietary intervention aims to provide adequate calories, protein, and other essential nutrients to maintain a healthy weight and preserve nutritional status during immunotherapy [59,60]. Additionally, (Enhancing Overall Well-Being), a nutritious diet can contribute to overall well-being by promoting energy levels, reducing fatigue, improving mood, and supporting mental health [61,62]. Breast cancer patients undergoing immunotherapy may experience physical and emotional challenges [63,64] and a healthy diet can play a role in supporting their overall quality of life (Figure 1).

### 4.5. Dietary HDAC2i

Compared to commonly used anti-cancer therapies, dietary interventions are safer and more cost-effective [65,66]. Even if dietary interventions cannot be considered a replacement for conventional cancer treatments, their role in improving the outcomes of cancer treatment also cannot be ignored [67,68] (Figure 3).

One advantage of dietary intervention is the minimal side effects compared to anti-cancer therapies [69,70]. Immunotherapy, chemotherapy, and radiation therapy can often cause adverse reactions such as fatigue, arthralgia, rash, pruritus, pneumonitis, acute kidney injury, and also a weakened immune system [71,72,73,74]. In contrast, dietary changes usually focus on incorporating natural, nutrient-rich foods, which are less likely to cause significant side effects [70].

Anti-cancer therapies often involve powerful and aggressive drugs that can be toxic to healthy and normal cells in the human body [75]. Such toxicity can lead to additional health complications and adversely affect a patient’s well-being [75]. Dietary interventions, on the other hand, prioritize whole foods, fruits, vegetables, and other plant-based sources that provide essential nutrients without the toxic effects associated with some medical treatments [76].

Cancer patients who undergo multiple treatments or take various medications simultaneously will face the increased risk of adverse effects due to drug interactions or unwanted chiral compounds [77]. Dietary interventions generally use natural foods rather than pharmacological compounds, which have both left- and right-hand chiral compounds in balance [78].

Dietary interventions can be customized to suit each individual’s needs and medical conditions [60]. This adaptability allows dietary plans to be tailored based on any pre-existing conditions, allergies, or specific dietary restrictions of each patient, ensuring a safer approach [60].

As a long-term lifestyle change, dietary interventions offer the advantage of being sustainable even after the initial treatment period. This sustainable approach helps maintain the patient’s overall health and reduces the risk of developing long-term side effects associated with certain anti-cancer therapies [79].

Dietary interventions can complement anti-cancer therapies by providing a supportive role [78]. A healthy diet can help strengthen the immune system, improve overall health, and enhance the body’s ability to cope with cancer treatments.

Certain dietary strategies, such as adopting a balanced and nutrient-rich diet, regular physical activity, and weight management, have the potential to reduce the risk of developing cancer [68,80,81,82,83,84]. Prevention is a crucial aspect of cancer management, and dietary interventions can also play a significant role in reducing the occurrence of cancer [78].

### 4.6. Selected Candidates of HDAC2i

Dietary compounds which possess HDAC2 inhibitory activity offer a new strategy for BC prevention and treatment. These potential candidates may enhance the anti-cancer effect of PD-1 inhibitors (Table 2).

#### 4.6.1. Genistein (GE)

GE is the most predominant bioactive isoflavone found mainly in soybean products and other food sources such as lupin, fava beans, kudzu, and psoralea [84]. GE has been reported to act as a potent chemo-preventive and therapeutic agent against various types of cancers including breast, prostate, and lung cancer [85]. For instance, GE suppresses the growth of BC cells in patient-derived tumor xenograft (PDX) [85]. Moreover, GE has been reported to modify the expression levels and activities of key epigenetic-associated genes, including HDAC2, DNA methyltransferases (DNMT3b) and ten-eleven translocation (TET3) methylcytosine dioxygenases. These genes are involved in epigenetic modifications, such as DNA methylation and histone methylation, which can regulate gene expression and impact cellular behavior [85]. By modulating the activities of these epigenetic regulators, GE may influence the epigenetic landscape of breast cancer cells, leading to changes in gene expression patterns that can affect cancer-related pathways.

#### 4.6.2. Sulforaphane (SFN)

SFN is a natural compound and is abundant in cruciferous vegetables such as broccoli sprouts (BSp) and kale [111]. It has gained attention for its potential health benefits, including its ability to inhibit HDAC2 activity [112,113]. In the context of breast cancer, a dietary regimen of genistein and BSp in combination has been shown effective in reducing mammary tumor incidence and delaying tumor latency in a spontaneous breast cancer mouse model [86]. The combination of GE and SFN downregulated HDAC2 protein levels in breast cancer cells. This suggests that the combined action of GE and SFN can influence gene expression in breast cancer cells by modulating HDAC2 activity, thereby affecting immune response [86].

#### 4.6.3. Chrysin and Its Analogues

Chrysin and its analogues are a group of polyphenolic compounds found in various dietary sources such as fruits, vegetables, olive oil, tea, and red wine [114]. The cytotoxic effects of chrysin have been shown against a wide range of cancer cell lines, including BC (MCF-7, MDA-MB-231), colon cancer (Lovo, DLD-1), and prostate cancer cells [87,88]. It is able to induce G1 cell cycle arrest and inhibit the activity of HDACs, specifically HDAC2 [89]. Moreover, polyphenolic compounds could promote the growth of SCFA-producer *Lachnoclostridium* [21], consequently modulating immune response.

#### 4.6.4. Resveratrol (RSV)

RSV is also a polyphenol abundant in grape skin and seeds. It also presents in other food sources such as apples, blueberries, mulberries, peanuts, pistachios, plums, and red wine [90]. RSV has numerous beneficial properties of anti-glycosylation, anti-inflammation, anti-neurodegeneration, and antioxidation in various types of cancer [91]. One intriguing aspect of RSV is its proposed potential as a pan-HDAC inhibitor [92]. Studies have shown that RSV can inhibit the growth of BC cells (MCF-7 and MDA-MB-231) by inhibiting the activity of HDAC2 in a dose-dependent manner [93].

#### 4.6.5. Oleuropein (OLE)

OLE is a polyphenolic compound in virgin olive oil with antineoplastic properties and it is well tolerated by humans [94]. Studies have shown that OLE can reduce progression, invasion, and proliferation of breast cancer cells by suppressing the activity of both HDAC2 and HDAC3 [95]. However, OLE exhibits little negative effect on normal breast epithelial cells, suggesting a potential selectivity towards BC cells and its potential for BC patients receiving ICIs therapy [95].

#### 4.6.6. Curcumin

Curcumin, a lipophilic polyphenol derived from turmeric (Curcuma longa), has been extensively studied for its diverse health-promoting properties, including antioxidant, anti-inflammatory, hepatoprotective, anti-atherosclerotic, and antidiabetic effects [96]. Moreover, curcumin has been investigated for its potential as an HDAC inhibitor in MCF-7 and MDA-MB-231 cells [97], showing inhibitive effects on both HDAC activity and the expression of HDAC 1 and 2 in a dose-dependent manner [97].

#### 4.6.7. Valeric Acid

Valerian (Valeriana officinalis) is a medicated diet that has been commonly used in cooking soup by some ethnic minorities in China for hundreds of years for restoring and balancing body energy [98]. Valeric acid, a major active component of valerian, has been identified as a potential HDAC inhibitor with anti-cancer effects on liver and breast cancer [98]. A study reported that valeric acid significantly decreases HDAC2 activity in treated breast cancer cells and may lead to alterations of DNA methylation [99].

#### 4.6.8. Rh4

Ginseng is also a typical medicated diet item and is commonly used for making cuisine mainly in Asia. It is also a traditional Chinese herb with multiple biological effects. One of its components, Rh4, has been identified as a rare ginsenoside with potential inhibitive effects on the development of various cancers [100]. Rh4 can inhibit the expression of PD-L1 by regulating HDAC2-mediated JAK/STAT in breast cancer cells [101]. In addition, a study of the binding of Rh4 and HDAC2 suggests a high binding affinity existing between Rh4 and HDAC2, indicating the potential of ginseng as a dietary intervention for BC patients [101].

#### 4.6.9. Butyrate (NaB)

NaB, a short-chain fatty acid generated via the fermentation of dietary fiber by the colonic microbiota, has shown anticancer activities mediated through HDACi [102]. NaB is primarily derived from undigested dietary carbohydrates, such as resistant starch and dietary fiber and, to a lesser extent, from dietary and endogenous proteins [103,104]. Studies have demonstrated that treatment with NaB, when combined with retinoids, enhances the inhibition of breast cancer cell proliferation [105]. Furthermore, the combination of butyrate with tumor necrosis factor-α (TNF-α), tumor necrosis factor-related apoptosis-inducing ligand (TRAIL), and anti-Fas agonist has been found to strongly induce apoptosis, leading to a significant decrease in the viability of breast cancer cells [106]. The action of NaB is often mediated through Sp1/Sp3 binding sites (e.g., p21 (Waf1/Cip1)). Both Sp1 and Sp3 were associated with HDAC activity in human breast cancer cells. And Sp1 and Sp3 recruit HDAC1 and HDAC2, with the latter being phosphorylated by protein kinase CK2 [115]. CK2 is upregulated in several cancers including breast cancer, which may promote breast cancer by deregulating key transcription processes [115].

#### 4.6.10. Other Potential Candidates

Some other dietary compounds have also been identified as HDAC2i in other cancer types. For instance, green tea and its bioactive components, especially polyphenols, possess many health-promoting and disease-preventing benefits with anti-inflammatory, antimutagenic, antioxidant, and anticancer properties, but have no significant toxicity on normal cells in vivo. It has the potential as an effective chemotherapeutic agent for cancer prevention and treatment through various cellular, molecular, and biochemical mechanisms [116]. The major polyphenol components of green tea are (-)-epigallocatechin-3-gallate (EGCG), (-)-epigallocatechin (EGC), (-)-epicatechin-3-gallate (ECG) and (-)-epicatechin (EC) [116]. One of the molecular mechanisms underlying the anticancer effects of green tea polyphenols (GTPs) is HDAC2 inhibition [107]. Another study presented GTPs suppressed melanoma via regulating the circ_MITF/miR-30e-3p/HDAC2 axis [108]. Rosmarinic acid (RA), a main phenolic compound in rosemary, presents anti-inflammatory, anti-oxidant, and anti-cancer effects [109]. RA induced cell cycle arrest and apoptosis through modulation of HDAC2 expression in prostate cancer [109]. In addition, ursolic acid (UA) is a well-known natural triterpenoid abundant in apple peels, basil (*Ocimum basilicum*), blueberries (*Vaccinium* spp.), cranberries (*Vaccinium macrocarpon*), heather flowers (*Calluna vulgaris*), Labrador tea (*Ledum groenlandicum* Retzius), olives (*Olea europaea*), pears (*Pyrus pyrifolia*), and rosemary (*Rosmarinus officinalis*). A study reported that UA reduced the expression of epigenetic modifying enzymes, including DNA methyltransferases DNMT1 and DNMT3a and histone deacetylases (HDACs) HDAC1, HDAC2, HDAC3 and HDAC8 (Class I), and HDAC6 and HDAC7 (Class II), and HDAC activity [110]. Given that fungal *Neurospora crassa* contains enriched retinal, and flavin adenine dinucleotide (FAD) or flavin mononucleotide (FMN), showing tumor growth inhibition of breast cancer [117,118], it will be interesting to explore whether *N. crassa* has potential as an adjuvant of ICIs. Overall, these dietary compounds mentioned above can also be considered as potential candidates. However, their HDAC2 inhibitory effects should be further confirmed and evaluated in breast cancer cells.

### 4.7. Potential Approaches of Taking Bioactive Compound

For enhancing absorption efficiency and therefore improving the potential health benefits and biological activities of certain bioactive compounds, many means of application have been developed [119]. Some of them might provide a better way for BC patients receiving ICIs to gain HDAC2i efficiently. However, it is important to emphasize that thorough exploration in this area is still a pressing necessity.

a.Nutraceuticals and Dietary Supplements

Many bioactive compounds with antioxidant or anti-inflammatory properties, found in certain fruits or vegetables, have been produced as nutraceuticals and dietary supplements for taking them more conveniently and easily [65]. Moreover, nutraceuticals and dietary supplements have been found to maintain excellent safety levels [120]. For instance, anthocyanins from berries, flavonols from dark chocolate, and resveratrol from red grapes have been widely used as consumed nutraceuticals [121]. Hence, dietary HDAC2i can be potentially produced as nutraceuticals and dietary supplements for clinical use.

b.Nanotechnology and Drug Delivery

Bioactive compounds can also be incorporated into well-designed nanoparticles for targeted drug delivery, enhancing drug efficacy and reducing side effects [122]. For instance, theracurmin, a curcumin formulation consisting of dispersed curcumin with colloidal nanoparticles, possesses significantly improved bioavailability and therapeutic efficacy for treating osteoarthritis, compared to turmeric powder monotherapy [123,124,125]. More specifically, by adding nanoparticles, theracurmin was shown to have greater bioavailability than turmeric powder by 40 fold in rats and 27 fold higher in humans [125], and to have fewer side effects [123]. Moreover, many nanoparticles have already been developed for specific targeted delivery to breast cancer cells with excellent safety, such as Cur-Dox-NPs (selective co-delivery of doxorubicin and curcumin), FeAC-DOX@PC-HCQ NPs, DHAPN, and Opaxio™ [126,127,128]. Thus, this approach has potential for widespread utilization among BC patients seeking HDAC2 inhibitors. Nevertheless, it is essential to emphasize that substantial research is imperative to substantiate its feasibility and efficacy.

c.Pharmaceuticals and Medicinal Products

Some bioactive compounds can be isolated and developed into pharmaceutical drugs to efficiently improve their therapeutic effect [129]. For instance, curcumin, a bioactive compound that has been found to possess multiple biological regulatory functions, has been successfully isolated from plant curcuma aromatica salisb for treating different types of cancer, including BC [130,131]. Moreover, paclitaxel, an efficient anti-cancer tricyclic diterpenoid compound, was also originally isolated from the plant Taxus brevifolia and subsequently synthesized for cancer treatment [132]. Therefore, plants containing HDAC2i may also be generated as pharmaceuticals and medicinal products.

d.Phytotherapy and Traditional Medicine

Phytotherapy and traditional medicine have been widely applied in treating various of diseases [133,134]. They are natural, with relatively low irritation and side effects on the human body, and can also be utilized in combination with other treatment [135,136]. Moreover, evidence has proven that these therapeutic approaches can help to enhance the therapeutic effect of anti-cancer treatment, such as chemotherapy [137]. Normally, patients can achieve certain active ingredients of nutrients by decocting herbal plants. For instance, valeric acid, the dietary HDAC2i mentioned above can be obtained by a traditional Chinese medicine decoction containing the valerian herb [98,99]. Thus, phytotherapy (or traditional medicine) seems to be a reliable way for patients to take HDAC2i. However, the effectiveness of these methods for absorbing HDAC2i needs more evaluation.

### 4.8. Nutrients That May Impair the Therapeutic Effect of ICIs

Even some dietary items that contain HDAC2i may improve the efficiency of ICIs, while other nutrients of diets that can potentially hamper the therapeutic effect of such therapy should also be noted [138]. Hence, those nutrients have been summarized below in order to emphasize the potential risks and provoke further exploration on their specific mechanisms and exact interactions.

a.Omega-3 Fatty Acids

Omega-3 fatty acids, commonly present in fish oil and certain plant sources, possess anti-inflammatory properties and are essential for synthesizing hormones and endogenous substances [139]. Natural killer (NK) cells are innate lymphocytes responsible for orchestrating immune responses against tumors and viruses [140]. Fish oil supplementation was found to decrease NK cell activity, which rebounded after supplementation ceased [141]. Notably, the age of individuals might influence the impact of omega-3 supplementation on NK cells [141]. Therefore, excessive consumption of omega-3 fatty acids has been considered to potentially hamper the normal function of immunity, which might further dampen the efficacy of ICIs. Thus, an appropriate amount of omega-3 intake is important for BC patients receiving ICIs. Of course, the exact mechanisms and interactions should be further explored.

b.
*Vitamins*


Vitamins are a type of trace organic substance obtained from food that can maintain normal physiological functions in humans [142]. Vitamins participate in the biochemical reactions of the human body and regulate metabolic functions, including immunity [143]. Deficiency or over intake of certain vitamins has been found to impair anti-cancer immunity, therefore affecting the efficiency of ICIs [138]. For instance, vitamin D has shown the ability to elevate the T-regulatory (Treg)/T-helper 17 (Th-17) cell ratio, leading to immune suppression and contributing to the onset of immune-related adverse events (irAEs), indicating its potential risk for patients receiving ICI therapy [144,145]. Moreover, vitamin A has also been reported to suppress the expression of PD-L1 causing cancer resistance to PD-1/PD-L1 blockade therapy [146,147]. In addition, evidence has shown that vitamin B6 can suppress PD-L1 expression and block the PD-1/PD-L1 signaling pathway [148]. Thus, extra attention is required when administrating vitamin supplementation for breast cancer patients receiving ICIs. More research is also indispensable in this field.

c.Probiotics

Probiotics, including bacteria and yeast, are living microorganisms [149,150]. Some of them have been commonly utilized to promote gut health, closely intertwined with immune function [149,150]. Recent evidence has newly pointed out that an excessive immune response in the gut induced by overconsumption of probiotics might constrain the systemic immune reaction necessary for the optimal efficacy of ICIs [151,152]. Briefly, a clinical study involving 46 melanoma patients indicated that taking over-the-counter probiotic supplements (for unrelated conditions) was linked to a 70% reduction in response rate to ICI treatment [152]. Therefore, probiotics should be approached cautiously in BC patients undergoing immunotherapy.

d.High-Fiber Diets

Fiber-rich diets primarily comprise two essential elements: soluble fiber and insoluble fiber. These vital components are found in an array of plant-based foods, including legumes, whole grains, cereals, vegetables, fruits, nuts, and seeds. Dietary fiber is composed of non-starch polysaccharides and various plant constituents like cellulose, resistant starch, and resistant dextrins [153]. High-fiber diets have been considered to modulate the gut microbiota and influence immune responses [154]. While a diverse gut microbiome is generally associated with better health, certain bacterial metabolites produced from high-fiber diets could potentially hamper the efficiency of ICIs [155]. More specifically, evidence has shown that, in non-small cell lung cancer patients, notably increased serum indoleamine-2,3-dioxygenase (IDO) levels, which were potentially produced by high-fiber diets, induced primary resistance to ICI treatment [155]. Thus, such bacterial metabolites might play a pivotal role in ICI resistance [155]. Therefore, the overall benefits of a high-fiber diet should be considered in balance.

e.Ketogenic diet

The ketogenic diet (KD) is characterized by high fat, low to moderate protein, and very low carbohydrate intake [156]. Evidence has shown that KD can lead to a downregulation of CTLA-4 and PD-1 expression on tumor-infiltrating lymphocytes (TILs), as well as PD-L1 expression on glioblastoma cells in animal models [157]. Also, it has been observed that the ketogenic diet KD can lead to the downregulation of cell membrane-associated PD-L1 [158]. Therefore, KD has the potential to reduce the effectiveness of PD-1/PD-L1 blockade therapy and should be avoided by patients using PD-1/PD-L1 inhibitors.

f.Protein-restricted diet

A low-protein diet serves as a therapeutic approach for managing inherited metabolic disorders like phenylketonuria and homocystinuria. Additionally, it can be employed in the treatment of kidney or liver ailments. Furthermore, a reduced intake of protein has been observed to potentially lower the risk of bone fractures, likely due to alterations in calcium [159]. Notably, recent studies have found that the deprivation of glutamine, a building block of proteins, can reduce PD-1 expression, indicating the potential to suppress the efficiency of PD-1 inhibitors [160].

## 5. Conclusions

According to recent data, the number of new cases of BC has accounted for about 11.7% of all malignant tumors. It has surpassed lung cancer as the most common malignant tumor, and its mortality rate has ranked among the top five of all malignant tumors [118]. In an effort to cure BC patients, ICIs (e.g., PD-1/PD-L1 inhibitors) have been introduced. However, it is still a challenge to provide durable and ideal clinical benefits and cure BC patients. Thus, novel ICI-based therapeutic strategies in combination have been studied to further improve the effect of treatment, such as HDAC2i plus a PD-1/PD-L1 inhibitor.

Dietary intervention, as a supportive therapy, is able to function as an HDAC2i through daily intake for cancer patients. An HDAC2i can suppress IFN-γ-induced PD-L1 expression and inhibit the process of PD-L1 nuclear translocation. Thus, a diet-derived HDAC2i has the potential to improve the clinical outcomes of BC patients, especially for those who are taking PD-1/PD-L1 inhibitors. Furthermore, to the best of our knowledge, there are still no relevant clinical guidelines on the application of an HDAC2i-containing diet for patients, especially for BC patients receiving ICIs. For such a group of patients, this novel dietary therapy can not only provide new dietary options but also may improve their clinical outcomes.

To sum up, multiple anti-cancer and immunomodulatory effects of HDAC2i guarantee future research into investigating such a novel dietary intervention for BC patients receiving ICIs. The potential of diet to enhance the therapeutic effect of ICIs is still to be fully evaluated. Certainly, further research is also required to identify more HDAC2i-containing foods for more dietary choices.

## Figures and Tables

**Figure 1 nutrients-15-03984-f001:**
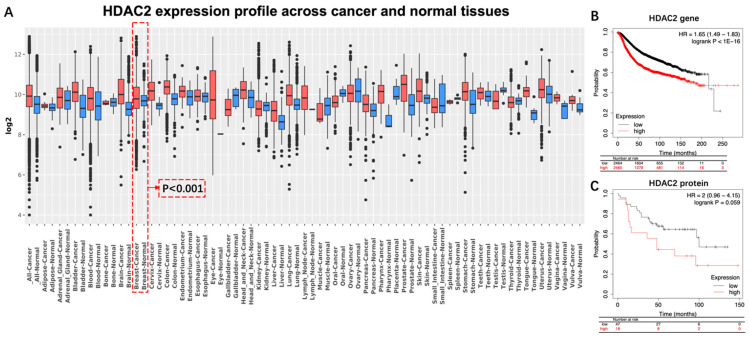
Differential expression and prognostic significance of HDAC2 in breast cancer. (**A**) Significant difference existed between normal breast tissue and breast cancer tissue (*p* < 0.001, Log2FC = 0.122): (**B**) BC patients with either high HDAC2 level of gene, or (**C**) high HDAC2 level of protein exhibited unfavorable overall survival. HR, hazard ratio; BC, breast cancer.

**Figure 2 nutrients-15-03984-f002:**
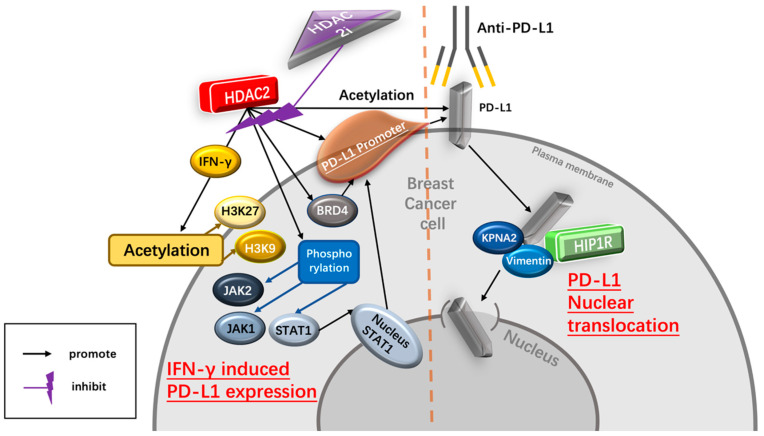
Potential mechanism of HDAC2i enhancing the therapeutic effect of the PD-1/PD-L1 inhibitor. HDAC2i suppresses IFNγ-induced PD-L1 expression regulated by HDAC2, therefore decreasing the immune escape of BC cells mediated by PD-L1. In addition, HDAC2i inhibits the process of PD-L1 nuclear translocation regulated by HDAC2/HIP1R axis, hence enhancing the therapeutic effect of PD-1 blockade treatment. Abbreviations: IFN-γ, interferon-γ; H3K27, histone 3 lysine 2; H3K9, histone 3 lysine 9; JAK2, Janus kinase 2; JAK1, Janus kinase 1; stat1, signal transducer and activator of transcription 1; BRD4, Bromodomain-containing protein 4; KPNA2, karyopherin α-2; HIP-1R, Huntingtin-interacting protein 1-related.

**Figure 3 nutrients-15-03984-f003:**
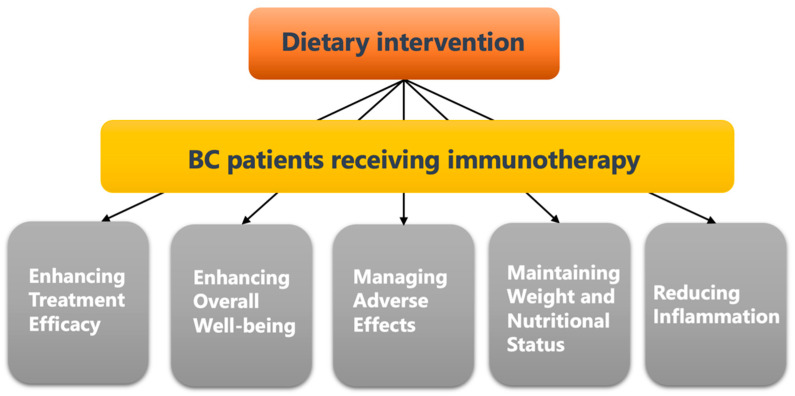
Potential advantages of applying proper dietary intervention in BC patients who receive ICIs treatment.

**Table 1 nutrients-15-03984-t001:** Correlation between HDAC2 protein expression and breast cancer.

Measurement	Sample Size	Patient Selection Criteria	Outcomes	Year	Refs
IHC	226	sporadic breast cancer patients who underwent surgery; All patients did not undergo radiation therapy and chemotherapy before surgery	High expression of HDAC2 was associated with: 1. advanced clinical stages (*p* = 0.016); 2. lymphatic metastasis (*p* = 0.02); 3. high histological grade (*p* = 0.001); 4. shorter OS of BC patients (*p* = 0.035); 5. shorter OS in multidrug resistance protein-positive patients (*p* = 0.034); 6. shorter survival in patients who received chemotherapy containing anthracyclines (OS, *p* = 0.041; disease-free survival, *p* = 0.084).	2016	[15]
IHC	118	Tumor size < 20 mm; Stage I, II and III; No preoperative anticancer therapy; Deceased from BC	High expression of HDAC2 was associated with: 1. BC in stage III (*p* < 0.001); 2. BC with histological grade 3 (*p* = 0.013).	2022	[13]
IHC	300	Invasive ductal carcinoma patients who underwent curative surgery	High expression of HDAC2 was correlated with improved OS in ER-negative BC patients (*p* = 0.048).	2014	[23]
IHC	212	patients with primary invasive breast cancer	High expression of HDAC2 was associated with: 1.overexpression of HER2 (*p* = 0.005); 2. lymphatic metastasis (*p* = 0.04).	2013	[24]

Abbreviations: BC, breast cancer; OS, overall survival; IHC, immunohistochemistry; DFS, disease-free survival.

**Table 2 nutrients-15-03984-t002:** Examples of dietary compounds identified as inhibiting HDAC2 activity.

Dietary Component	Food Source	Potential Benefits in Breast Cancer	Diseases	Refs
Genistein (GE)	Soybean products	Anti-cancer effects on breast cancer, modulation of Dnmt3b, Tet3 and HDAC	Breast cancer, cervical cancer,	[85]
Sulforaphane (SFN)	Broccoli sprouts, kale	Cytotoxic effects in breast, colon, and prostate cancer cells, inhibition of HDAC-2 and HDAC-8	Breast cancer, colon cancer, prostate cancer, neurodegenerative diseases, bladder carcinoma,	[86]
Chrysin	Fruits, vegetables, olive oil and red wine	Cytotoxic effects in breast, colon, and prostate cancer cells, inhibition of HDAC-2 and HDAC-8	Breast cancer, melanoma, fibrosarcoma, leukemia	[87,88,89]
Resveratrol (RSV)	Grapes, apples, blueberries, mulberries, peanuts, pistachios, plums, and red wine	Induced ATP2A3 upregulation correlates with reduced HDAC activity and reduced nuclear HDAC2 expression and occupancy on ATP2A3 promoter	Breast cancer, glioma	[90,91,92,93]
Oleuropein (OLE)	Virgin olive oil	Antineoplastic properties, modulation of HDAC2 and HDAC3 in breast cancer cells	Breast cancer, alzheimer,	[94,95]
Curcumin	Turmeric	Inhibited both HDAC activity and the expression of HDACs 1 and 2 in a concentration-dependent manner in cancer cells	Breast Cancer, lung cancer, hematological cancers, inflammation, arthritis, metabolic syndrome,	[96,97]
Valeric acid	Valerian herb	Anti-cancer effects on liver and breast cancer, modulation of HDAC2 and HDAC3	Breast cancer, prostate cancer, liver cancer	[98,99]
Ginsenoside Rh4	Ginseng herb	Inhibition of PD-L1 expression by regulating HDAC2-mediated JAK/STAT pathway in breast cancer cells	Breast cancer, lung adenocarcinoma, colorectal cancer, gastric cancer	[100,101]
Butyrate	Dietary fiber, resistant starch, undigested carbohydrates	Anticancer activity, inhibition of breast cancer cell proliferation, induction of apoptosis	Multiple cancer, cardiovascular disease, and type 2 diabetes	[102,103,104,105,106]
Green tea polyphenols	Green tea	Suppressed cancer progression by regulating circ_MITF/miR-30e-3p/HDAC2 axis	breast cancer, malignant melanoma, prostate cancer,	[107,108]
Rosmarinic acid (RA)	Rosemary tea	Potential pan-HDAC inhibitor, inhibition of nuclear HDAC2 protein levels in breast cancer cells	Breast cancer, prostate cancer	[109]
Ursolic acid (UA)	Blueberries, cranberries and apple peels	Reduced the expression of epigenetic modifying enzymes, including the DNMT1 and DNMT3a and the histone deacetylases (HDACs) HDAC1, HDAC2, HDAC3, HDAC6 and HDAC7 activity	Skin cancer, breast cancer, colorectal cancer	[110]

## Data Availability

Not applicable.

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
