# Peer review of "Potential of Dietary HDAC2i in Breast Cancer Patients Receiving PD-1/PD-L1 Inhibitors"

_nutrients, 2023, doi:10.3390/nu15183984_

Round 1
Reviewer 1 Report
The authors showed that the increase in Histone deacetylases 2 (HDAC 2) is involved with a worse prognosis in breast cancer (BC). Thus, HDAC2 inhibitors and PD-1/PD-L1 inhibitors could benefit patients with BC, however there are side effects. The authors suggest interaction with dietary intervention to improve the use of such medications.
Major corrections:
Describe the search methodology for articles, such as keywords used, databases used, etc.
A healthy lifestyle, based on the use of foods rich in fiber, polyphenols, etc., is really beneficial for the normal body and with any disease. However, we must be very careful about making generalized statements, especially when the individual is undergoing treatment for breast cancer. Studies have reported that some nutrients (especially when given as supplements) can interact with and reduce the effect of chemotherapy drugs such as tamoxifen and others. I suggest adding a topic emphasizing this situation. Authors should also reflect: How would the use of bioactive compounds be? Only those from food? As a supplement? How would the absorption of these compounds be to improve the side effects of immunotherapies?
Minor corrections:
Abstract: Lines 28-30: “This article not only provides a whole new therapeutic strategy of HDAC2i containing diet combined with PD-1/PD-L1 inhibitors for BC treatment, also aims to ignite enthusiasm to explore such field.” introduce the word "but", or a synonym, before "also aims"
Keywords: Usually, keywords other than those present in the title are used, as this increases the search for your article. I suggest changing the repeated words, such as "Dietary intervention; Breast Cancer".
Lines 98-99: “HDAC2 expression compared to breast- 98 cancer tissue and (P<0.001, Log2FC=0.122)>..)” Remove “and” because there is no words ahead. The same for the caption of figure 1.
Figure 2: Was figure 2 created using a specific program that should be mentioned in the caption? For example: Biorender
Author Response
Dear editor and reviewer,
Thank you so much for your consideration of our work, offering an opportunity to allow us to submit the revised manuscript. We also want to extend our appreciation to the reviewers for their critical comments and thoughtful suggestions. We have carefully and thoroughly revised the manuscript following each single comment, and addressed the concerns point- by-point raised by the reviewers as follows. We hope our revision is satisfactory.
Should any additional revisions are necessary, we would be happy to revise it further.
Thank you again.
By all authors
1
The authors showed that the increase in Histone deacetylases 2 (HDAC 2) is involved with a worse prognosis in breast cancer (BC). Thus, HDAC2 inhibitors and PD-1/PD-L1 inhibitors could benefit patients with BC, however there are side effects. The authors suggest interaction with dietary intervention to improve the use of such medications.
Major corrections:
(1)Describe the search methodology for articles, such as keywords used, databases used, etc.
Answer:
We thank the reviewer for his/her great comment. We have carefully and thoroughly described the details of methods in the section of “Materials and methods” as required, shown as following:
“The comprehensive literature search was conducted to identify relevant reports from 2000 to 2023, using PubMed and Web of Science databases. The search utilized various keywords, either individually or in combinations, including PD1, PDL1, breast cancer, HDAC2 inhibitor, immune checkpoint inhibitors, dietary intervention, herbal remedy, neoadjuvant, and immunotherapy sensitizer. The identified articles were thoroughly reviewed, with preference given to the most recent ones. Additional sources were extracted from the reference lists of selected papers. Priority was primarily assigned to original research and review articles based on animal studies and clinical trials.
Differential expression and prognostic analysis. The differential expression analysis of HDAC2 across cancer and normal tissues were obtained from GENT2 platform (http://gent2.appex.kr/gent2/). The prognostic analysis was performed by applying Kaplan‑Meier plotter platform (https://kmplot.com/analysis/index.php?p=background). BC patients with high or low HDAC2 protein (or gene) expression were divided by median expression level or by best cutoff value. Cutoff values used in analysis for HDAC2 gene and protein, were 1568 and 3, respectfully. The total amount of samples for analysing HDAC2 gene figure is 4929 (high 2465 vs. low 2464), its counterpart for HDAC2 protein is 65 (high 47 vs. low 18).”(line 105-121)
(2)A healthy lifestyle, based on the use of foods rich in fiber, polyphenols, etc., is really beneficial for the normal body and with any disease. However, we must be very careful about making generalized statements, especially when the individual is undergoing treatment for breast cancer. Studies have reported that some nutrients (especially when given as supplements) can interact with and reduce the effect of chemotherapy drugs such as tamoxifen and others. I suggest adding a topic emphasizing this situation.
Answer:
Thank you for your very professional comment. We are totally agreed with the opinions of our reviewer. Therefore, we added a topic to emphasize the situation mentioned above, titled as “Nutrients that may impair the therapeutic effect of immune checkpoint inhibitors”, shown as following:
“4.8 Nutrients that may impair the therapeutic effect of ICIs Dietary HDAC2i
Even some dietary that contains HDAC2i may improve the efficiency of ICIs, other nutrients of diets that can potentially hamper the therapeutic effect of such therapy should also be noticed[23]. Hence, those nutrients have been summarized below for emphasizing the potential risks and provoking further exploration on their specific mechanisms and exact interactions.
a.Omega-3 Fatty Acids
Omega-3 fatty acids, commonly present in fish oil and certain plant sources, possess anti-inflammatory properties and are essential for synthesizing hormones and endogenous substances[24]. Natural killer (NK) cells are innate lymphocytes responsible for orchestrating immune responses against tumors and viruses [25]. Fish oil supplementation was found to decrease NK cell activity, which rebounded after supplementation ceased[26]. Notably, the age of individuals might influence the impact of omega-3 supplementation on NK cells [26]. Therefore, excessive consumption of omega-3 fatty acids has been considered to potentially hamper the normal function of immunity, which might further dampen the efficacy of ICIs. Thus, an appropriate amount of omega-3 intake is important for BC patients receiving ICIs. Of course, the exact mechanisms and interactions should be further explored.
b.Vitamins
Vitamins are a type of trace organic substances obtained from food that can maintain normal physiological functions in humans[27]. Vitamins participate in the biochemical reactions of the human body and regulate metabolic functions, including immunity [28]. Deficiency or over intake of certain vitamin has been found to impair the anti-cancer immunity therefore affecting the efficiency of ICIs[23]. For instance, vitamin D has shown the ability to elevate the T-regulatory (Treg)/T-helper 17 (Th-17) cell ratio, leading to immune suppression and contributing to the onset of immune-related adverse events (irAEs), indicating its potential risk for patients receiving ICIs therapy[29, 30]. Moreover, Vitamin A has also been reported to suppress the expression of PD-L1 causing cancer resistance to PD-1/PD-L1 blockade therapy [31, 32]. In addition, evidence have found that Vitamin B6 can Suppress PD-L1 expression and block the PD-1/PD-L1 signaling pathway [33]. Thus, extra attention is required when administrating vitamin supplementation for breast cancer patients receiving ICIs. More research is also indispensable for such field.
c.Probiotics
Probiotics, including bacteria and yeast, are live microorganisms[34, 35]. Some of them have been commonly utilized to promote gut health, closely intertwined with immune function[34, 35]. Recent evidence has newly pointed out that an excessive immune response in the gut induced by overconsumption of probiotics might constrain the systemic immune reaction necessary for the optimal efficacy of ICIs[36] [37]. Briefly, a clinical study involving 46 melanoma patients indicated that taking over-the-counter probiotic supplements (for unrelated conditions) was linked to a 70% reduction in response rate to ICI treatment[37]. Therefore, the probiotics should be approached cautiously in BC patients undergoing immunotherapy.
d.High-Fiber Diets
Fiber-rich diets primarily comprise two essential elements: soluble fiber and insoluble fiber. These vital components are found in an array of plant-based foods, including legumes, whole grains, cereals, vegetables, fruits, nuts, and seeds. Dietary fiber is composed of non-starch polysaccharides and various plant constituents like cellulose, resistant starch, and resistant dextrins[38]. High-Fiber Diets have been considered to modulate the gut microbiota and influence immune responses[39]. While a diverse gut microbiome is generally associated with better health, certain bacterial metabolites produced from high-fiber diets could potentially hamper the efficiency of ICIs[40]. For details, evidence displayed that, in non-small cell lung cancer patients, notably increased serum indoleamine-2,3-dioxygenase (IDO) levels which was potentially produced by high-fiber diets, has induced the primary resistance to ICI treatment [40]. Thus, such bacterial metabolite might play a pivotal role in ICI resistance[40]. Therefore, the overall benefits of a high-fiber diet should be considered in balance.
e.Ketogenic diet
The ketogenic diet (KD) is characterized by high fat, low to moderate protein, and very low carbohydrate intake [41]. evidence has shown that KD can lead to a downregulation of CTLA-4 and PD-1 expression on tumor-infiltrating lymphocytes (TILs), as well as PD-L1 expression on glioblastoma cells in animal models [42]. Also, it has been observed that the ketogenic diet KD can lead to the downregulation of cell membrane-associated PD-L1[43]. Therefore, KD has the potential to reduce the effectiveness of PD-1/PD-L1 blockade therapy and should be avoided by patients using PD-1/PD-L1 inhibitors.
f.Protein restricted diet
A low-protein diet serves as a therapeutic approach for managing inherited metabolic disorders like phenylketonuria and homocystinuria. Additionally, it can be employed in the treatment of kidney or liver ailments. Furthermore, a reduced intake of protein has been observed to potentially lower the risk of bone fractures, likely due to alterations in calcium[44]. Notably, Novel evidence has found that the deprivation of glutamine, a building block of proteins, can reduce PD-1 expression, indicating the potential to suppress the efficiency of PD-1 inhibitors[45].”(line 606- line 666)
(3)Authors should also reflect: How would the use of bioactive compounds be? Only those from food? As a supplement? How would the absorption of these compounds be to improve the side effects of immunotherapies?
Answer:
Thank you for your very professional comments. Based on the excellent suggestion provided by our reviewer, we have now added an extra section to further discuss the potential applying means of HDAC2i, for providing more future research directions and revealing more potential of diet HDAC2i, shown as following:
“4.7 Potential approaches of taking bioactive compound
For enhancing the absorption efficiency therefore improving the potential health benefits and biological activities of certain bioactive compound, many applying means have been developed[1]. Some of them might provide a better way for BC patients receiving ICIs to gaining HDAC2i efficiently. However, it's important to emphasize that thorough exploration in this area is still a pressing necessity.
a.Nutraceuticals and Dietary Supplements
Many bioactive compounds with antioxidant or anti-inflammatory properties, found in certain fruits or vegetables, have been produced as nutraceuticals and dietary supplements for taking them more convenient and easily[2]. Moreover, nutraceuticals and dietary supplements have been found to hold excellent safety[3]. For instance, anthocyanins from berries, flavonols from dark chocolate and resveratrol from red grapes have been widely used as consumed nutraceuticals[4]. Hence, dietary HDAC2i can be potentially produced as nutraceuticals and dietary supplements for clinical use.
b.Nanotechnology and Drug Delivery
Bioactive compounds can also be incorporated into well designed nanoparticles for targeted drug delivery, enhancing drug efficacy and reducing side effects [5]. For instance, theracurmin, a curcumin formulation consisting of dispersed curcumin with colloidal nanoparticles, possesses significantly improved bioavailability and therapeutic efficacy for treating osteoarthritis, compared to turmeric powder monotherapy [6-8]. For details, by adding nanoparticles , theracurmin was shown to have greater bioavailability than turmeric powder by 40 fold in rats and 27 fold higher in humans [8], and have less side effects [6]. Moreover, many nanoparticles have already been developed for breast cancer cells specific targeted delivery with excellent safety, such as Cur-Dox-NPs (selective co-delivery of doxorubicin and curcumin), FeAC-DOX@PC-HCQ NPs, DHAPN and Opaxio™ [9-11]. Thus, this approach holds the potential for widespread utilization among BC patients seeking HDAC2 inhibitors. Nevertheless, it's essential to emphasize that substantial research is imperative to substantiate its feasibility and efficacy.
c.Pharmaceuticals and Medicinal Products
Some bioactive compounds can be isolated and developed into pharmaceutical drugs for efficiently improving their therapeutic effect[12]. For instance, curcumin, a bioactive compound that has been found to possess multiple biological regulatory functions, has been successfully isolated from plant curcuma aromatica salisb.,for treating different types of cancer including BC[13, 14]. Moreover, paclitaxel, an efficient anti-cancer tricyclic diterpenoid compound, was also originally isolated plant Taxus brevifolia., and subsequently synthesized for cancer treatment[15]. Therefore, plants containing HDAC2i may also be generated as pharmaceuticals and medicinal products.
d.Phytotherapy and Traditional Medicine
Phytotherapy and Traditional Medicine have been widely applied in treating various of diseases[16, 17]. They are natural, with relatively low irritation and side effects on the human body, also can be utilized in combination with other treatment[18, 19]. Moreover, evidence has proved that these therapeutic approaches can help to enhance the therapeutic effect of anti-cancer treatment, such as chemotherapy[20]. Normally, patients can achieve certain active ingredients of nutrients by decocting herbal plants. For instance, valeric acid, the dietary HDAC2i mentioned above can be gained by a traditional Chinese medicine decoction containing valerian herb [21, 22]. Thus, phytotherapy (or traditional Medicine) seems to be a reliable way for patients to taking HDAC2i. However, the effectiveness of these methods for absorbing HDAC2i needs more evaluation.” (line 561- line 605)
References for section 4.7 and 4.8
- Dima, C., et al., Oral bioavailability of bioactive compounds; modulating factors, in vitro analysis methods, and enhancing strategies. Crit Rev Food Sci Nutr, 2023: p. 1-39.
- Martínez-Garay, C. and N. Djouder, Dietary interventions and precision nutrition in cancer therapy. Trends Mol Med, 2023. 29(7): p. 489-511.
- Rosenfeld, R.M., H.M. Juszczak, and M.A. Wong, Scoping review of the association of plant-based diet quality with health outcomes. Front Nutr, 2023. 10: p. 1211535.
- Weaver, C.M., et al., Flavonoid intake and bone health. J Nutr Gerontol Geriatr, 2012. 31(3): p. 239-53.
- Li, B., et al., Nano-drug co-delivery system of natural active ingredients and chemotherapy drugs for cancer treatment: a review. Drug Deliv, 2022. 29(1): p. 2130-2161.
- Nakagawa, Y., et al., Short-term effects of highly-bioavailable curcumin for treating knee osteoarthritis: a randomized, double-blind, placebo-controlled prospective study. J Orthop Sci, 2014. 19(6): p. 933-9.
- Kanai, M., et al., Dose-escalation and pharmacokinetic study of nanoparticle curcumin, a potential anticancer agent with improved bioavailability, in healthy human volunteers. Cancer Chemother Pharmacol, 2012. 69(1): p. 65-70.
- Sasaki, H., et al., Innovative preparation of curcumin for improved oral bioavailability. Biol Pharm Bull, 2011. 34(5): p. 660-5.
- Gao, C., et al., pH-Responsive prodrug nanoparticles based on a sodium alginate derivative for selective co-release of doxorubicin and curcumin into tumor cells. Nanoscale, 2017. 9(34): p. 12533-12542.
- Zhang, H., et al., Co-delivery of doxorubicin and hydroxychloroquine via chitosan/alginate nanoparticles for blocking autophagy and enhancing chemotherapy in breast cancer therapy. Front Pharmacol, 2023. 14: p. 1176232.
- Dong, X., et al., Synergistic Combination of Bioactive Hydroxyapatite Nanoparticles and the Chemotherapeutic Doxorubicin to Overcome Tumor Multidrug Resistance. Small, 2021. 17(18): p. e2007672.
- Li, Z., et al., Drug delivery for bioactive polysaccharides to improve their drug-like properties and curative efficacy. Drug Deliv, 2017. 24(sup1): p. 70-80.
- Prasad, S., et al., Curcumin, a component of golden spice: from bedside to bench and back. Biotechnol Adv, 2014. 32(6): p. 1053-64.
- Passos, C.L.A., et al., Curcumin and melphalan cotreatment induces cell cycle arrest and apoptosis in MDA-MB-231 breast cancer cells. Sci Rep, 2023. 13(1): p. 13446.
- Zhu, L. and L. Chen, Progress in research on paclitaxel and tumor immunotherapy. Cell Mol Biol Lett, 2019. 24: p. 40.
- Nootim, P., et al., Current state of cancer patient care incorporating Thai traditional medicine in Thailand: A qualitative study. J Integr Med, 2020. 18(1): p. 41-45.
- Yazdi, N., et al., Use of complementary and alternative medicine in pregnant women: A cross-sectional survey in the south of Iran. J Integr Med, 2019. 17(6): p. 392-395.
- Ouyang, W., et al., Efficacy and safety of traditional Chinese medicine in the treatment of osteonecrosis of the femoral head. J Orthop Surg Res, 2023. 18(1): p. 600.
- Bu, Z.J., et al., Comparative effectiveness and safety of Chinese medicine belly button application for childhood diarrhea: a Bayesian network meta-analysis of randomized controlled trials. Front Pediatr, 2023. 11: p. 1180694.
- Hemmati Bushehri, R., et al., Integration of phytotherapy and chemotherapy: Recent advances in anticancer molecular pathways. Iran J Basic Med Sci, 2023. 26(9): p. 987-1000.
- Shi, F., et al., Valerian and valeric acid inhibit growth of breast cancer cells possibly by mediating epigenetic modifications. Sci Rep, 2021. 11(1): p. 2519.
- Han, R., et al., Valeric acid acts as a novel HDAC3 inhibitor against prostate cancer. Med Oncol, 2022. 39(12): p. 213.
- Zhang, X., et al., Impact of Diets on Response to Immune Checkpoint Inhibitors (ICIs) Therapy against Tumors.Life (Basel), 2022. 12(3).
- Kromhout, D., et al., Fish oil and omega-3 fatty acids in cardiovascular disease: do they really work? Eur Heart J, 2012. 33(4): p. 436-43.
- Björkström, N.K., B. Strunz, and H.G. Ljunggren, Natural killer cells in antiviral immunity. Nat Rev Immunol, 2022. 22(2): p. 112-123.
- Thies, F., et al., Dietary supplementation with eicosapentaenoic acid, but not with other long-chain n-3 or n-6 polyunsaturated fatty acids, decreases natural killer cell activity in healthy subjects aged >55 y. Am J Clin Nutr, 2001. 73(3): p. 539-48.
- Fortmann, S.P., et al., Vitamin and mineral supplements in the primary prevention of cardiovascular disease and cancer: An updated systematic evidence review for the U.S. Preventive Services Task Force. Ann Intern Med, 2013. 159(12): p. 824-34.
- Beck, K.L., et al., Micronutrients and athletic performance: A review. Food Chem Toxicol, 2021. 158: p. 112618.
- Daniel, C., et al., Immune modulatory treatment of trinitrobenzene sulfonic acid colitis with calcitriol is associated with a change of a T helper (Th) 1/Th17 to a Th2 and regulatory T cell profile. J Pharmacol Exp Ther, 2008. 324(1): p. 23-33.
- Larkin, J., et al., Combined Nivolumab and Ipilimumab or Monotherapy in Untreated Melanoma. N Engl J Med, 2015. 373(1): p. 23-34.
- Tobin, R.P., et al., Targeting myeloid-derived suppressor cells using all-trans retinoic acid in melanoma patients treated with Ipilimumab. Int Immunopharmacol, 2018. 63: p. 282-291.
- Chen, L., et al., CD38-Mediated Immunosuppression as a Mechanism of Tumor Cell Escape from PD-1/PD-L1 Blockade. Cancer Discov, 2018. 8(9): p. 1156-1175.
- Yuan, J., et al., Identification of vitamin B6 as a PD-L1 suppressor and an adjuvant for cancer immunotherapy.Biochem Biophys Res Commun, 2021. 561: p. 187-194.
- Kim, S.K., et al., Role of Probiotics in Human Gut Microbiome-Associated Diseases. J Microbiol Biotechnol, 2019. 29(9): p. 1335-1340.
- Legesse Bedada, T., et al., Probiotics for cancer alternative prevention and treatment. Biomed Pharmacother, 2020. 129: p. 110409.
- Suez, J., et al., Post-Antibiotic Gut Mucosal Microbiome Reconstitution Is Impaired by Probiotics and Improved by Autologous FMT. Cell, 2018. 174(6): p. 1406-1423.e16.
- Spencer, C.N., et al., Abstract 2838: The gut microbiome (GM) and immunotherapy response are influenced by host lifestyle factors. Cancer Research, 2019. 79(13_Supplement): p. 2838-2838.
- Kuang, R. and D.G. Binion, Should high-fiber diets be recommended for patients with inflammatory bowel disease? Curr Opin Gastroenterol, 2022. 38(2): p. 168-172.
- Marques, F.Z., et al., High-Fiber Diet and Acetate Supplementation Change the Gut Microbiota and Prevent the Development of Hypertension and Heart Failure in Hypertensive Mice. Circulation, 2017. 135(10): p. 964-977.
- Kocher, F., et al., High indoleamine-2,3-dioxygenase 1 (IDO) activity is linked to primary resistance to immunotherapy in non-small cell lung cancer (NSCLC). Transl Lung Cancer Res, 2021. 10(1): p. 304-313.
- Weber, D.D., et al., Ketogenic diet in the treatment of cancer - Where do we stand? Mol Metab, 2020. 33: p. 102-121.
- Lussier, D.M., et al., Enhanced immunity in a mouse model of malignant glioma is mediated by a therapeutic ketogenic diet. BMC Cancer, 2016. 16: p. 310.
- Rom-Jurek, E.M., et al., Regulation of Programmed Death Ligand 1 (PD-L1) Expression in Breast Cancer Cell Lines In Vitro and in Immunodeficient and Humanized Tumor Mice. Int J Mol Sci, 2018. 19(2).
- Orillion, A., et al., Dietary Protein Restriction Reprograms Tumor-Associated Macrophages and Enhances Immunotherapy. Clin Cancer Res, 2018. 24(24): p. 6383-6395.
- Nabe, S., et al., Reinforce the antitumor activity of CD8(+) T cells via glutamine restriction. Cancer Sci, 2018. 109(12): p. 3737-3750.
Minor corrections:
Abstract: Lines 28-30: “This article not only provides a whole new therapeutic strategy of HDAC2i containing diet combined with PD-1/PD-L1 inhibitors for BC treatment, also aims to ignite enthusiasm to explore such field.” introduce the word "but", or a synonym, before "also aims"
Answer:
Thank the reviewer for his/her great comment. We have now revised the sentence as: “This article not only provides a whole new therapeutic strategy of HDAC2i containing diet combined with PD-1/PD-L1 inhibitors for BC treatment, but also aims to ignite enthusiasm to explore such field.”(Line 28-30), as required.
Keywords: Usually, keywords other than those present in the title are used, as this increases the search for your article. I suggest changing the repeated words, such as "Dietary intervention; Breast Cancer".
Answer:
Thank you so much for your excellent suggestion. It really helps us to improve the quality of this manuscript. We have now replaced “Dietary intervention”, and “Breast Cancer", by “Dietotherapy”, “Breast carcinoma”, respectively (Line 31).
Lines 98-99: “HDAC2 expression compared to breast- 98 cancer tissue and (P<0.001, Log2FC=0.122)>..)” Remove “and” because there is no words ahead. The same for the caption of figure 1.
Answer:
Thank you so much for your very thoughtful comment. We have revised the sentences as: “we found that normal breast tissue had a significant lower HDAC2 expression compared to breast-cancer tissue (P<0.001, Log2FC=0.122)(Fig.1A)”(line 124-125), and “Differential expression and prognostic significance of HDAC2 in breast cancer. Significant difference existed between breast-normal tissue and breast-cancer tissue (P<0.001, Log2FC=0.122)(A)”(caption of figure 1).
Figure 2: Was figure 2 created using a specific program that should be mentioned in the caption? For example: Biorender
Answer:
Thank you so much for your very important reminder. The original figure of potential mechanisms (figure 2) was created by using normal program (PPT, Microsoft powerpoint version16.53), just like those mechanism diagrams in our published articles, such as: (https://doi.org/10.3390/cancers15030824)( DOI 10.3389/fimmu.2023.1052657).

Reviewer 2 Report
In the manuscript a (literature) overview on specific treatments of breast cancer patients is presented (for patients receiving specific inhibitors with HDAC). The authors seem to be experts in this field and are going deep into details. So this paper could be used as an interesting substantiated background paper for medical doctors and developer of therapies. The focus is lying on dietary support of breast cancer patients.
The common structure of a scientific paper (Introduction, Material and Methods, Results, Discussion) is missing (though it starts with “1. Introduction”, but no 2... et cetera follows) and, therefore, the manuscript does not meet the usual requirements of a scientific work.
It becomes not clear for what proportion of breast cancer patients all the considerations made are relevant and for which patient group this is applicable at all. The authors wrote “the benefit population of breast cancer from ICI monotherapy is limited”, but this remains without explanation.
Contrary to my expectation (triggered by the title) the authors are not presenting an own study. So all the presented information are coming from other studies and/or other groups. This is documented by referring to a large number of literature (119 references!).
It is nearly not possible to find out, where the information presented in Figures 1 to 3 is coming from, because no sources are mentioned. Especially for Figure 1A the reader wants to know where these data for this graphic presentation are coming from and how many patients are building the basis. The reference to a GENT-webpage seems inappropriate. In Figure 2 abbreviations are not explained. Why contains Table 1 only four studies [15,13,23,24], though in the accompanying text a fifth one [14] is mentioned?
The title of the paper seems misleading to me (see above). Independently of this, the wording “to support breast cancer patients” seems an unusual expression to me.
Overall, I recommend to reject this manuscript because (a) the common structure of scientific publications is not existent (b) no Methods are described (c) sources of data (mainly in the Figures) are not named (d) numbers of patients (or portions) of breast cancer patients considered are not given (e) the relevance for breast cancer patients in general is not appropriately visible.Author Response
Dear editor and reviewer,
Thank you so much for your consideration of our work, offering an opportunity to allow us to submit the revised manuscript. We also want to extend our appreciation to the reviewers for their critical comments and thoughtful suggestions. We have carefully and thoroughly revised the manuscript following each single comment, and addressed the concerns point- by-point raised by the reviewers as follows. We hope our revision is satisfactory.
Should any additional revisions are necessary, we would be happy to revise it further.
Thank you again.
By all authors
2
In the manuscript a (literature) overview on specific treatments of breast cancer patients is presented (for patients receiving specific inhibitors with HDAC). The authors seem to be experts in this field and are going deep into details. So this paper could be used as an interesting substantiated background paper for medical doctors and developer of therapies. The focus is lying on dietary support of breast cancer patients.
(1)The common structure of a scientific paper (Introduction, Material and Methods, Results, Discussion) is missing (though it starts with “1. Introduction”, but no 2... et cetera follows) and, therefore, the manuscript does not meet the usual requirements of a scientific work.
Answer:
Thank the reviewer for his/her very specific comments and very important reminder. We are deeply sorry for the issue of paper structure. The original version of this manuscript did have those parts mentioned above. However, we separated those content into each section for better logical presentation.
Now, based on the reviewer’s requirements, also according to the instruction for authors of 《Nutrients》(required sections: Author Information, Abstract, Keywords, Introduction, Materials & Methods, Results, Conclusions, Figures and Tables with Captions, Funding Information…)(https://www.mdpi.com/journal/nutrients/instructions), we have thoroughly and carefully reorganized our manuscript to meet the structure requirements strictly:
“1.Introdution”(Line 34-100)
“2. Materials and Methods:
The comprehensive literature search was conducted to identify relevant reports from 2000 to 2023, using PubMed and Web of Science databases. The search utilized various keywords, either individually or in combinations, including PD1, PDL1, breast cancer, HDAC2 inhibitor, immune checkpoint inhibitors, dietary intervention, herbal remedy, neoadjuvant, and immunotherapy sensitizer. The identified articles were thoroughly reviewed, with preference given to the most recent ones. Additional sources were extracted from the reference lists of selected papers. Priority was primarily assigned to original research and review articles based on animal studies and clinical trials.
Differential expression and prognostic analysis. The differential expression analysis of HDAC2 across cancer and normal tissues were obtained from GENT2 platform (http://gent2.appex.kr/gent2/). The prognostic analysis was performed by applying Kaplan‑Meier plotter platform (https://kmplot.com/analysis/index.php?p=background). BC patients with high or low HDAC2 protein (or gene) expression were divided by median expression level or by best cutoff value. Cutoff values used in analysis for HDAC2 gene and protein, were 1568 and 3, respectfully. The total amount of samples for analysing HDAC2 gene figure is 4929 (high 2465 vs. low 2464), its counterpart for HDAC2 protein is 65.”(line 105-121)
“3.results:
HDAC2 expression profile across cancer and normal tissues has been detected by using GENT2 platform. Significant difference in HDAC2 expression has been displayed between breast-cancer tissue and breast-normal tissue (P<0.001, Log2FC,0.122)(Fig.1A)(Significant test results by Two-sample T-test for HDAC2 expression profile has been displayed in supplementary materials); As shown in Kaplan‑Meier survival curves (Fig.1B and 1C), BC Patients with high expression of gene HDAC2 [hazard ratio (HR), 1.65; P<1E-16; Fig. 1B] or high expression level of HDAC2 protein (HR, 2; P=0.059; Fig. 1C) exhibited a less favorable prognosis compared with patients with low expression level. However, the significant difference has not been detected in the group of HDAC2 protein.” (line 127-136);
“4. Discussion” (line 199-666);
“5. Conclusion”.
Moreover, we also completed the numerical order of all titles in the manuscript:
“1. Introduction;
2. Materials and Methods;
3. Results ;
4. Discussion;
4.1 HDAC2, a potential index of aggressiveness and a therapeutic target against BC;
4.2 HDAC2 inhibition for treating breast cancer;
4.3 HDAC2 inhibition enhances the therapeutic effect of ICIs in BC treatment.;
4.3.1 HDAC2 regulates PD-L1 nuclear translocation;
4.3.2 HDAC2 regulates IFN-γ- induced PD-L1 expression;
a. IFN-γ upregulates the expression of PD-L1 ;
b. HDAC2i affects IFN-γ induced PD-L1 ;
4.4 Dietary intervention is important for breast cancer patients receiving anti-cancer immunotherapy ;
4.5 Dietary HDAC2i ;
4.6 Selected candidates of HDAC2i;
4.7 Potential approaches of taking bioactive compound;
4.8 Nutrients that may impair the therapeutic effect of ICIs;
5. Conclusion”.
(2)It becomes not clear for what proportion of breast cancer patients all the considerations made are relevant and for which patient group this is applicable at all. The authors wrote “the benefit population of breast cancer from ICI monotherapy is limited”, but this remains without explanation.
Answer:
Thank you for your great comments and questions.
For explaining this, the limited clinical benefit of ICI monotherapy is a common clinical issue and an objective fact(doi: 10.1007/978-3-319-70197-4_10). As described in (doi: 10.1007/978-3-319-70197-4_10): “Immune checkpoint inhibition has demonstrated modest single agent activity in advanced breast cancer. While overall response rates are relatively low”.( doi: 10.1016/j.surg.2019.09.018): “Given the somewhat limited efficacy of ICI monotherapy in breast cancer, much interest has been focused on combining ICI and other therapeutic modalities, including conventional chemotherapy, other targeted therapies, radiation, and cancer vaccines”. (doi.org/10.3390/cancers15030824): “the therapeutic effect of PD-1/PD-L1 inhibitor monotherapy for breast cancer is limited, resulting in more attention on the multiple immune checkpoint blockade therapeutic strategy [62,63]”. Also as described in (doi: 10.3389/fonc.2020.600573):“To conclude, although the response rates of single agent ICIs in mTNBC may be modest, the durable responses of a subset of PD-L1 positive patients suggest that combination treatment of immune checkpoint blockade with other treatment modalities may provide a favorable outcome”. “the results of these studies suggested that ICI monotherapy could offer limited survival benefit, potentially enhanced in patients with PD-L1+ breast cancer”(DOI: 10.1200/OP.22.00483 JCO Oncology).
The benefit population for breast cancer patients from ICI monotherapy is limited for reasons like Tumor Heterogeneity, Immunologically "Cold" Tumors, Lack of Biomarkers, Resistance Mechanisms, etc(DOI: 10.1200/OP.22.00483 JCO Oncology)( doi: 10.1016/j.surg.2019.09.018).
Based on the evidence that “the benefit population of breast cancer from ICI monotherapy is limited”, combination therapy was developed and designed for further enhancing the therapeutic effect. A number of randomized clinical trials of ICIs in combination with chemotherapy, targeted therapies, radiotherapy, and other therapeutic agents are underway, and many evidence has displayed that combinations can boost the overall immune response and improve the chances of treatment success, compared to ICI monotherapy (doi: 10.1007/978-3-319-70197-4_10). For instance, In the phase 2 I-SPY 2 (NCT01042379) study the addition of pembrolizumab to taxane- and anthracycline-based neoadjuvant chemotherapy doubled the estimated pathological complete response (pCR) rates of early stage patients with Her2-negative breast cancer including triple negative breast cancer (doi: 10.1001/jamaoncol.2019.6650). Moreover, the phase 1b clinical study NCT01633970 reported an ORR of 39.4% with a median PFS of 5.5 months for locally advanced or metastatic TNBC patients treated with atezolizumab plus nab-paclitaxel, compared to atezolizumab monotherapy group where an ORR of 24% and median PFS of 1.6 months was observed (doi: 10.1001/jamaoncol.2018.5152).
(3)Contrary to my expectation (triggered by the title) the authors are not presenting an own study. So all the presented information are coming from other studies and/or other groups. This is documented by referring to a large number of literature (119 references!).
Answer:
Thank you for your comments. We are sorry for the potential confusion that might be caused by our title. Therefore, we are here to clarify that this article is actually a review paper which was meant to be a survey of previously published research on a topic. To collect recent progress in the particular topic of this article, we have to present information from other studies and other groups, just like other published papers did. As for our relevant studies, such as citation [103] and [104], have already been discussed in manuscript: “Valerian (Valeriana officinalis) is a medicated diet that has been commonly used in cooking soup by some ethnic minorities in China for hundreds of years for restoring and balancing body energy [103]. Valeric acid, a major active component of Valerian, has been identified as a potential HDAC inhibitor with anti-cancer effects on liver and breast cancer[103]. A study reported that valeric acid significantly decreases HDAC2 activity in treated breast cancer cells and may lead to alterations of DNA methylation [104]”. The reason we didn’t present the figures or data of our published articles, is due to the copyright issue. Moreover, as a reviewer paper, it is unreasonable and impossible to only included authors’ own research. It is apparently against the purpose of review article.
Furthermore, the large number of our cited literatures, can actually indicate that the evidence we collected to support our topic is strong and comprehensive. For instance, one of our recently published review articles (doi.org/10.3390/ cancers15030824) in 《Cancers》(also belongs to MDPI) contains 178 citations. Even our mini-review paper (doi:10.3389/fimmu.2023.1052657) recently published in 《Frontiers in immunology》(IF:8) possesses 62 literature. Moreover, even a random review paper from 《Cancer cell》(IF:50.3) (https://doi.org/10.1016/j.ccell.2021.08.006) contains 300 references. We are sorry we haven’t spotted any problems for the issue of reference number.
(4)It is nearly not possible to find out, where the information presented in Figures 1 to 3 is coming from, because no sources are mentioned. Especially for Figure 1A the reader wants to know where these data for this graphic presentation are coming from and how many patients are building the basis. The reference to a GENT-webpage seems inappropriate.
Answer:
Thank you for your professional comments. the information source of Figures 1 has been added in the section of Materials and Methods: “Differential expression and prognostic analysis. The differential expression analysis of HDAC2 across cancer and normal tissues were obtained from GENT2 platform (http://gent2.appex.kr/gent2/). The prognostic analysis was performed by applying Kaplan‑Meier plotter platform (https://kmplot.com/analysis/index.php?p=background). BC patients with high or low HDAC2 protein (or gene) expression were divided by median expression level or by best cutoff value. Cutoff values used in analysis for HDAC2 gene and protein, were 1568 and 3, respectfully.” As for figure 3, a summarized flow chart, the information was summarized from citation of 69-84. Which has been thoroughly and comprehensively discussed in the section of “4.5 dietary HDAC2i” (Line 399-442).
For Figure 1A, the information of details has been now added in manuscript:” “Differential expression and prognostic analysis. The differential expression analysis of HDAC2 across cancer and normal tissues were obtained from GENT2 platform (http://gent2.appex.kr/gent2/)”, “HDAC2 expression profile across cancer and normal tissues has been detected by using GENT2 platform. Significant difference in HDAC2 expression has been displayed between breast-cancer tissue and breast-normal tissue (P<0.001, Log2FC,0.122)(Fig.1A)(Significant test results by Two-sample T-test for HDAC2 expression profile has been displayed in supplementary materials)”(Line 127-132). As for the patient number, the GENT2 platform never provides such data, however, the detail information for KMP analysis has also been added: “Kaplan‑Meier plotter platform (https://kmplot.com/analysis/index.php?p=background). BC patients with high or low HDAC2 protein (or gene) expression were divided by median expression level or by best cutoff value. Cutoff values used in analysis for HDAC2 gene and protein, were 1568 and 3, respectfully. The total amount of samples for analysing HDAC2 gene figure is 4929 (high 2465 vs. low 2464), its counterpart for HDAC2 protein is 65 (high 47 vs. low 18). ”(Line 116-121)
The GENT2 analysis combined with KMP analysis were conducted to further displayed the correlations between HDAC2 and BC, they are both necessary to support our topic. Just as mentioned in the section of “4.1 HDAC2, a potential index of aggressiveness and a therapeutic target against BC”: “Those results are also consistent with the outcomes from our analysis. For instance, our results displayed that expression of HDAC2 in BC tissue was significantly higher than that in breast-normal tissue (P<0.001)(Fig.1); Moreover, the trend can be observed that BC patients with high expression of HDAC2 gained worse prognosis than those are not(Fig.1C).” (Line 251-255)
(5)In Figure 2 abbreviations are not explained. Why contains Table 1 only four studies [15,13,23,24], though in the accompanying text a fifth one [14] is mentioned?
Answer:
Thank you for such helpful comment. We have now added abbreviations at the end of figure 2 legends: “Figure 2. Potential mechanism of HDAC2i enhancing the therapeutic effect of PD-1/PD-L1 inhibitor. HDAC2i suppresses IFNγ-induced PD-L1 expression regulated by HDAC2, therefore decreasing the immune escape of BC cells mediated by PD-L1. In addition, HDAC2i inhibits the process of PD-L1 nuclear translocation regulated by HDAC2/HIP1R axis, hence enhancing the therapeutic effect of PD-1 blockade treatment. Abbreviations—IFN-γ, interferon-γ; H3K27, histone 3 lysine 2; H3K9, Histone 3 lysine 9; JAK2, Janus kinase 2; JAK1, Janus kinase 1; stat1, signal transducer and activator of transcription 1; BRD4, Bromodomain-containing protein 4; KPNA2, karyopherin α-2; HIP-1R, Huntingtin-interacting protein 1-related. ” (Line 310-317)
(6)The title of the paper seems misleading to me (see above). Independently of this, the wording “to support breast cancer patients” seems an unusual expression to me.
Answer:
Thank you for such great comment. The fifth reference (Shan, W., et al., HDAC2 overexpression correlates with aggressive clinicopathological features and DNA-damage response pathway of breast cancer. Am J Cancer Res, 2017. 7(5): p. 1213-1226. PMCID: PMC5446485) is a in vitro assay. The result of this article is consistent with the outcomes of clinical studies selected in table.1 that overexpression of HDAC2 is significantly correlated with high tumor grade, positive lymph node status, and poor prognosis of breast cancer. Therefore, this refence has been cited in the accompanying text to collectively clarify the correlations between HDAC2 expression and BC.
(7)Overall, I recommend to reject this manuscript because (a) the common structure of scientific publications is not existent (b) no Methods are described (c) sources of data (mainly in the Figures) are not named (d) numbers of patients (or portions) of breast cancer patients considered are not given (e) the relevance for breast cancer patients in general is not appropriately visible.
Answer:
Thank you for such professional comment. (a) we have now thoroughly revised the structure, and also added numerical order of all titles in the manuscript, as shown in the first answer; (b) “Materials & Methods” has now been added (line 101-121), as shown in the first answer; (c) Sources of data (mainly in the Figures) have been added, as responded in answer 4; (d) Numbers of patients (or portions) of breast cancer patients considered have been added as required, as shown in answer 4. (e) This manuscript has been developed and revised strictly based on substantial evidence from both preclinical and clinical studies of breast cancer patients. Moreover, the revision has added and revised more content to apparently discuss the relevance for breast cancer patients. Thus, current version can further display the visibility of relevance mentioned by reviewer, collectively shown in answer 1-6.

Reviewer 3 Report
The first part of this scientific paper dedicated to genetics and epigenetics in breast cancer therapy is not my area of expertise.
The part of the paper that refers to potential nutritional candidates for inclusion in the diet of breast cancer patients includes a rather concise review of the commonly known components of the anticancer diet.
My assessment is that there are no new insights, it is only about systematizing the previous knowledge, at least as far as the dietary part of this work is concerned.
Author Response
Dear editor and reviewer,
Thank you so much for your consideration of our work, offering an opportunity to allow us to submit the revised manuscript. We also want to extend our appreciation to the reviewers for their critical comments and thoughtful suggestions. We have carefully and thoroughly revised the manuscript following each single comment, and addressed the concerns point- by-point raised by the reviewers as follows. We hope our revision is satisfactory.
Should any additional revisions are necessary, we would be happy to revise it further.
Thank you again.
By all authors
3
The first part of this scientific paper dedicated to genetics and epigenetics in breast cancer therapy is not my area of expertise.
The part of the paper that refers to potential nutritional candidates for inclusion in the diet of breast cancer patients includes a rather concise review of the commonly known components of the anticancer diet.
My assessment is that there are no new insights, it is only about systematizing the previous knowledge, at least as far as the dietary part of this work is concerned.
Answer:
Thank you for your professional comment. We are so sorry that the novelty of this manuscript has been badly neglected.
Accumulating evidence from our team and other research groups summarized in this article, has newly reveal the synergistic effect of HDAC2 inhibition in ICIs therapy. In this manuscript, we are the first to comprehensively summarize and discuss other evidence that concerns BC patients.
More importantly, more than just being “only about systematizing the previous knowledge”, to our best knowledge, this is the first manuscript which points out and discusses the potential of dietary containing HDAC2i in breast cancer patients who are receiving PD-1/PD-L1 inhibitors, and there is still no relevant clinical guideline recommending the application of an HDAC2i-containing diet for this type of patients. For our oncology physicians, such novel dietothrapy not only can provide new dietary option, and also may actually improve the clinical outcomes of BC patient receiving ICIs therapy. Why there were no new insights?
Such whole new dietary intervention strategy with great potential for BC cancers treatment is firstly brought about by this article, and built on the accumulated and published studies about HDAC2 of our group, combined with other latest research. As physicians of oncology department, our team members thought such potential strategy contains massive value for further exploration in BC treatment, which might improve the clinical benefits if more attention and energy has been invested into this whole new area.
In addition, as a review paper, one of the purposes is to comprehensively collect and summarize existing evidence. Even for the dietary part, collecting previous knowledge is necessary. Moreover, new evidence for dietary part, such as Valeric acid acting as HDAC2i recently published by our team (reference 103: Han, R., et al., Valeric acid acts as a novel HDAC3 inhibitor against prostate cancer. Med Oncol, 2022. 39(12): p. 213; Reference 104. Shi, F., et al., Valerian and valeric acid inhibit growth of breast cancer cells possibly by mediating epigenetic modifications. Sci Rep, 2021. 11(1): p. 2519.) has also been concluded. Therefore, the summarization of diet containing HDAC2i is also updated and more comprehensive. Also, the importance of finding a wise way to apply nutrient equals reporting a new one. We have also added two sections “4.7 Potential approaches of taking bioactive compound, 4.8 Nutrients that may impair the therapeutic effect of ICIs” (Line 561-666) to further discuss the potential application of dietary HDAC2i for BC patients receiving ICIs therapy
For further indicating the novelty and significance of this manuscript. We added content in “conclusion”: “Furthermore, to the best of our knowledge, there are still no relevant clinical guidelines on the application of an HDAC2i-containing diet for patients, especially for BC patients receiving ICIs. For such a group of patients, this novel dietary therapy can not only provide new dietary options but also may improve their clinical outcomes”(Line 682-686).
The newly added sections of 4.7 and 4.8 have been shown below:
4.7 Potential approaches of taking bioactive compound
For enhancing the absorption efficiency therefore improving the potential health benefits and biological activities of certain bioactive compound, many applying means have been developed[1]. Some of them might provide a better way for BC patients receiving ICIs to gaining HDAC2i efficiently. However, it's important to emphasize that thorough exploration in this area is still a pressing necessity.
a.Nutraceuticals and Dietary Supplements
Many bioactive compounds with antioxidant or anti-inflammatory properties, found in certain fruits or vegetables, have been produced as nutraceuticals and dietary supplements for taking them more convenient and easily[2]. Moreover, nutraceuticals and dietary supplements have been found to hold excellent safety[3]. For instance, anthocyanins from berries, flavonols from dark chocolate and resveratrol from red grapes have been widely used as consumed nutraceuticals[4]. Hence, dietary HDAC2i can be potentially produced as nutraceuticals and dietary supplements for clinical use.
b.Nanotechnology and Drug Delivery
Bioactive compounds can also be incorporated into well designed nanoparticles for targeted drug delivery, enhancing drug efficacy and reducing side effects [5]. For instance, theracurmin, a curcumin formulation consisting of dispersed curcumin with colloidal nanoparticles, possesses significantly improved bioavailability and therapeutic efficacy for treating osteoarthritis, compared to turmeric powder monotherapy [6-8]. For details, by adding nanoparticles , theracurmin was shown to have greater bioavailability than turmeric powder by 40 fold in rats and 27 fold higher in humans [8], and have less side effects [6]. Moreover, many nanoparticles have already been developed for breast cancer cells specific targeted delivery with excellent safety, such as Cur-Dox-NPs (selective co-delivery of doxorubicin and curcumin), FeAC-DOX@PC-HCQ NPs, DHAPN and Opaxio™ [9-11]. Thus, this approach holds the potential for widespread utilization among BC patients seeking HDAC2 inhibitors. Nevertheless, it's essential to emphasize that substantial research is imperative to substantiate its feasibility and efficacy.
c.Pharmaceuticals and Medicinal Products
Some bioactive compounds can be isolated and developed into pharmaceutical drugs for efficiently improving their therapeutic effect[12]. For instance, curcumin, a bioactive compound that has been found to possess multiple biological regulatory functions, has been successfully isolated from plant curcuma aromatica salisb.,for treating different types of cancer including BC[13, 14]. Moreover, paclitaxel, an efficient anti-cancer tricyclic diterpenoid compound, was also originally isolated plant Taxus brevifolia., and subsequently synthesized for cancer treatment[15]. Therefore, plants containing HDAC2i may also be generated as pharmaceuticals and medicinal products.
d.Phytotherapy and Traditional Medicine
Phytotherapy and Traditional Medicine have been widely applied in treating various of diseases[16, 17]. They are natural, with relatively low irritation and side effects on the human body, also can be utilized in combination with other treatment[18, 19]. Moreover, evidence has proved that these therapeutic approaches can help to enhance the therapeutic effect of anti-cancer treatment, such as chemotherapy[20]. Normally, patients can achieve certain active ingredients of nutrients by decocting herbal plants. For instance, valeric acid, the dietary HDAC2i mentioned above can be gained by a traditional Chinese medicine decoction containing valerian herb [21, 22]. Thus, phytotherapy (or traditional Medicine) seems to be a reliable way for patients to taking HDAC2i. However, the effectiveness of these methods for absorbing HDAC2i needs more evaluation.” (line 561- line 605)
4.8 Nutrients that may impair the therapeutic effect of ICIs Dietary HDAC2i
Even some dietary that contains HDAC2i may improve the efficiency of ICIs, other nutrients of diets that can potentially hamper the therapeutic effect of such therapy should also be noticed[23]. Hence, those nutrients have been summarized below for emphasizing the potential risks and provoking further exploration on their specific mechanisms and exact interactions.
a.Omega-3 Fatty Acids
Omega-3 fatty acids, commonly present in fish oil and certain plant sources, possess anti-inflammatory properties and are essential for synthesizing hormones and endogenous substances[24]. Natural killer (NK) cells are innate lymphocytes responsible for orchestrating immune responses against tumors and viruses [25]. Fish oil supplementation was found to decrease NK cell activity, which rebounded after supplementation ceased[26]. Notably, the age of individuals might influence the impact of omega-3 supplementation on NK cells [26]. Therefore, excessive consumption of omega-3 fatty acids has been considered to potentially hamper the normal function of immunity, which might further dampen the efficacy of ICIs. Thus, an appropriate amount of omega-3 intake is important for BC patients receiving ICIs. Of course, the exact mechanisms and interactions should be further explored.
b.Vitamins
Vitamins are a type of trace organic substances obtained from food that can maintain normal physiological functions in humans[27]. Vitamins participate in the biochemical reactions of the human body and regulate metabolic functions, including immunity [28]. Deficiency or over intake of certain vitamin has been found to impair the anti-cancer immunity therefore affecting the efficiency of ICIs[23]. For instance, vitamin D has shown the ability to elevate the T-regulatory (Treg)/T-helper 17 (Th-17) cell ratio, leading to immune suppression and contributing to the onset of immune-related adverse events (irAEs), indicating its potential risk for patients receiving ICIs therapy[29, 30]. Moreover, Vitamin A has also been reported to suppress the expression of PD-L1 causing cancer resistance to PD-1/PD-L1 blockade therapy [31, 32]. In addition, evidence have found that Vitamin B6 can Suppress PD-L1 expression and block the PD-1/PD-L1 signaling pathway [33]. Thus, extra attention is required when administrating vitamin supplementation for breast cancer patients receiving ICIs. More research is also indispensable for such field.
c.Probiotics
Probiotics, including bacteria and yeast, are live microorganisms[34, 35]. Some of them have been commonly utilized to promote gut health, closely intertwined with immune function[34, 35]. Recent evidence has newly pointed out that an excessive immune response in the gut induced by overconsumption of probiotics might constrain the systemic immune reaction necessary for the optimal efficacy of ICIs[36] [37]. Briefly, a clinical study involving 46 melanoma patients indicated that taking over-the-counter probiotic supplements (for unrelated conditions) was linked to a 70% reduction in response rate to ICI treatment[37]. Therefore, the probiotics should be approached cautiously in BC patients undergoing immunotherapy.
d.High-Fiber Diets
Fiber-rich diets primarily comprise two essential elements: soluble fiber and insoluble fiber. These vital components are found in an array of plant-based foods, including legumes, whole grains, cereals, vegetables, fruits, nuts, and seeds. Dietary fiber is composed of non-starch polysaccharides and various plant constituents like cellulose, resistant starch, and resistant dextrins[38]. High-Fiber Diets have been considered to modulate the gut microbiota and influence immune responses[39]. While a diverse gut microbiome is generally associated with better health, certain bacterial metabolites produced from high-fiber diets could potentially hamper the efficiency of ICIs[40]. For details, evidence displayed that, in non-small cell lung cancer patients, notably increased serum indoleamine-2,3-dioxygenase (IDO) levels which was potentially produced by high-fiber diets, has induced the primary resistance to ICI treatment [40]. Thus, such bacterial metabolite might play a pivotal role in ICI resistance[40]. Therefore, the overall benefits of a high-fiber diet should be considered in balance.
e.Ketogenic diet
The ketogenic diet (KD) is characterized by high fat, low to moderate protein, and very low carbohydrate intake [41]. evidence has shown that KD can lead to a downregulation of CTLA-4 and PD-1 expression on tumor-infiltrating lymphocytes (TILs), as well as PD-L1 expression on glioblastoma cells in animal models [42]. Also, it has been observed that the ketogenic diet KD can lead to the downregulation of cell membrane-associated PD-L1[43]. Therefore, KD has the potential to reduce the effectiveness of PD-1/PD-L1 blockade therapy and should be avoided by patients using PD-1/PD-L1 inhibitors.
f.Protein restricted diet
A low-protein diet serves as a therapeutic approach for managing inherited metabolic disorders like phenylketonuria and homocystinuria. Additionally, it can be employed in the treatment of kidney or liver ailments. Furthermore, a reduced intake of protein has been observed to potentially lower the risk of bone fractures, likely due to alterations in calcium[44]. Notably, Novel evidence has found that the deprivation of glutamine, a building block of proteins, can reduce PD-1 expression, indicating the potential to suppress the efficiency of PD-1 inhibitors[45].”(line 606- line 666)
References for section 4.7 and 4.8
- Dima, C., et al., Oral bioavailability of bioactive compounds; modulating factors, in vitro analysis methods, and enhancing strategies. Crit Rev Food Sci Nutr, 2023: p. 1-39.
- Martínez-Garay, C. and N. Djouder, Dietary interventions and precision nutrition in cancer therapy. Trends Mol Med, 2023. 29(7): p. 489-511.
- Rosenfeld, R.M., H.M. Juszczak, and M.A. Wong, Scoping review of the association of plant-based diet quality with health outcomes. Front Nutr, 2023. 10: p. 1211535.
- Weaver, C.M., et al., Flavonoid intake and bone health. J Nutr Gerontol Geriatr, 2012. 31(3): p. 239-53.
- Li, B., et al., Nano-drug co-delivery system of natural active ingredients and chemotherapy drugs for cancer treatment: a review. Drug Deliv, 2022. 29(1): p. 2130-2161.
- Nakagawa, Y., et al., Short-term effects of highly-bioavailable curcumin for treating knee osteoarthritis: a randomized, double-blind, placebo-controlled prospective study. J Orthop Sci, 2014. 19(6): p. 933-9.
- Kanai, M., et al., Dose-escalation and pharmacokinetic study of nanoparticle curcumin, a potential anticancer agent with improved bioavailability, in healthy human volunteers. Cancer Chemother Pharmacol, 2012. 69(1): p. 65-70.
- Sasaki, H., et al., Innovative preparation of curcumin for improved oral bioavailability. Biol Pharm Bull, 2011. 34(5): p. 660-5.
- Gao, C., et al., pH-Responsive prodrug nanoparticles based on a sodium alginate derivative for selective co-release of doxorubicin and curcumin into tumor cells. Nanoscale, 2017. 9(34): p. 12533-12542.
- Zhang, H., et al., Co-delivery of doxorubicin and hydroxychloroquine via chitosan/alginate nanoparticles for blocking autophagy and enhancing chemotherapy in breast cancer therapy. Front Pharmacol, 2023. 14: p. 1176232.
- Dong, X., et al., Synergistic Combination of Bioactive Hydroxyapatite Nanoparticles and the Chemotherapeutic Doxorubicin to Overcome Tumor Multidrug Resistance. Small, 2021. 17(18): p. e2007672.
- Li, Z., et al., Drug delivery for bioactive polysaccharides to improve their drug-like properties and curative efficacy. Drug Deliv, 2017. 24(sup1): p. 70-80.
- Prasad, S., et al., Curcumin, a component of golden spice: from bedside to bench and back. Biotechnol Adv, 2014. 32(6): p. 1053-64.
- Passos, C.L.A., et al., Curcumin and melphalan cotreatment induces cell cycle arrest and apoptosis in MDA-MB-231 breast cancer cells. Sci Rep, 2023. 13(1): p. 13446.
- Zhu, L. and L. Chen, Progress in research on paclitaxel and tumor immunotherapy. Cell Mol Biol Lett, 2019. 24: p. 40.
- Nootim, P., et al., Current state of cancer patient care incorporating Thai traditional medicine in Thailand: A qualitative study. J Integr Med, 2020. 18(1): p. 41-45.
- Yazdi, N., et al., Use of complementary and alternative medicine in pregnant women: A cross-sectional survey in the south of Iran. J Integr Med, 2019. 17(6): p. 392-395.
- Ouyang, W., et al., Efficacy and safety of traditional Chinese medicine in the treatment of osteonecrosis of the femoral head. J Orthop Surg Res, 2023. 18(1): p. 600.
- Bu, Z.J., et al., Comparative effectiveness and safety of Chinese medicine belly button application for childhood diarrhea: a Bayesian network meta-analysis of randomized controlled trials. Front Pediatr, 2023. 11: p. 1180694.
- Hemmati Bushehri, R., et al., Integration of phytotherapy and chemotherapy: Recent advances in anticancer molecular pathways. Iran J Basic Med Sci, 2023. 26(9): p. 987-1000.
- Shi, F., et al., Valerian and valeric acid inhibit growth of breast cancer cells possibly by mediating epigenetic modifications. Sci Rep, 2021. 11(1): p. 2519.
- Han, R., et al., Valeric acid acts as a novel HDAC3 inhibitor against prostate cancer. Med Oncol, 2022. 39(12): p. 213.
- Zhang, X., et al., Impact of Diets on Response to Immune Checkpoint Inhibitors (ICIs) Therapy against Tumors.Life (Basel), 2022. 12(3).
- Kromhout, D., et al., Fish oil and omega-3 fatty acids in cardiovascular disease: do they really work? Eur Heart J, 2012. 33(4): p. 436-43.
- Björkström, N.K., B. Strunz, and H.G. Ljunggren, Natural killer cells in antiviral immunity. Nat Rev Immunol, 2022. 22(2): p. 112-123.
- Thies, F., et al., Dietary supplementation with eicosapentaenoic acid, but not with other long-chain n-3 or n-6 polyunsaturated fatty acids, decreases natural killer cell activity in healthy subjects aged >55 y. Am J Clin Nutr, 2001. 73(3): p. 539-48.
- Fortmann, S.P., et al., Vitamin and mineral supplements in the primary prevention of cardiovascular disease and cancer: An updated systematic evidence review for the U.S. Preventive Services Task Force. Ann Intern Med, 2013. 159(12): p. 824-34.
- Beck, K.L., et al., Micronutrients and athletic performance: A review. Food Chem Toxicol, 2021. 158: p. 112618.
- Daniel, C., et al., Immune modulatory treatment of trinitrobenzene sulfonic acid colitis with calcitriol is associated with a change of a T helper (Th) 1/Th17 to a Th2 and regulatory T cell profile. J Pharmacol Exp Ther, 2008. 324(1): p. 23-33.
- Larkin, J., et al., Combined Nivolumab and Ipilimumab or Monotherapy in Untreated Melanoma. N Engl J Med, 2015. 373(1): p. 23-34.
- Tobin, R.P., et al., Targeting myeloid-derived suppressor cells using all-trans retinoic acid in melanoma patients treated with Ipilimumab. Int Immunopharmacol, 2018. 63: p. 282-291.
- Chen, L., et al., CD38-Mediated Immunosuppression as a Mechanism of Tumor Cell Escape from PD-1/PD-L1 Blockade. Cancer Discov, 2018. 8(9): p. 1156-1175.
- Yuan, J., et al., Identification of vitamin B6 as a PD-L1 suppressor and an adjuvant for cancer immunotherapy.Biochem Biophys Res Commun, 2021. 561: p. 187-194.
- Kim, S.K., et al., Role of Probiotics in Human Gut Microbiome-Associated Diseases. J Microbiol Biotechnol, 2019. 29(9): p. 1335-1340.
- Legesse Bedada, T., et al., Probiotics for cancer alternative prevention and treatment. Biomed Pharmacother, 2020. 129: p. 110409.
- Suez, J., et al., Post-Antibiotic Gut Mucosal Microbiome Reconstitution Is Impaired by Probiotics and Improved by Autologous FMT. Cell, 2018. 174(6): p. 1406-1423.e16.
- Spencer, C.N., et al., Abstract 2838: The gut microbiome (GM) and immunotherapy response are influenced by host lifestyle factors. Cancer Research, 2019. 79(13_Supplement): p. 2838-2838.
- Kuang, R. and D.G. Binion, Should high-fiber diets be recommended for patients with inflammatory bowel disease? Curr Opin Gastroenterol, 2022. 38(2): p. 168-172.
- Marques, F.Z., et al., High-Fiber Diet and Acetate Supplementation Change the Gut Microbiota and Prevent the Development of Hypertension and Heart Failure in Hypertensive Mice. Circulation, 2017. 135(10): p. 964-977.
- Kocher, F., et al., High indoleamine-2,3-dioxygenase 1 (IDO) activity is linked to primary resistance to immunotherapy in non-small cell lung cancer (NSCLC). Transl Lung Cancer Res, 2021. 10(1): p. 304-313.
- Weber, D.D., et al., Ketogenic diet in the treatment of cancer - Where do we stand? Mol Metab, 2020. 33: p. 102-121.
- Lussier, D.M., et al., Enhanced immunity in a mouse model of malignant glioma is mediated by a therapeutic ketogenic diet. BMC Cancer, 2016. 16: p. 310.
- Rom-Jurek, E.M., et al., Regulation of Programmed Death Ligand 1 (PD-L1) Expression in Breast Cancer Cell Lines In Vitro and in Immunodeficient and Humanized Tumor Mice. Int J Mol Sci, 2018. 19(2).
- Orillion, A., et al., Dietary Protein Restriction Reprograms Tumor-Associated Macrophages and Enhances Immunotherapy. Clin Cancer Res, 2018. 24(24): p. 6383-6395.
- Nabe, S., et al., Reinforce the antitumor activity of CD8(+) T cells via glutamine restriction. Cancer Sci, 2018. 109(12): p. 3737-3750.

Round 2
Reviewer 2 Report
Everything is fine now.
Reviewer 3 Report
I am satisfied with the explanations that the authors sent and I accept the paper.